# Type I interferon drives T cell cytotoxicity by upregulation of interferon regulatory factor 7 in autoimmune kidney diseases in mice

Huiying Wang [1,2], Jonas Engesser[1,2], Robin Khatri [2,3], Darius P. Schaub [2,3], Hans-Joachim Paust[1,2], Zeba Sultana[1,2,3], Saskia-Larissa Jauch-Speer [1], Anett Peters[1], Anna Kaffke[1], Stefan Bonn [2,3,4], Tobias B. Huber [1,2,4], Hans-Willi Mittrücker [2,5], Christian F. Krebs [1,2,4], Ulf Panzer [1,2,4] ✉ & Nariaki Asada [1,2] ✉

In anti-neutrophil cytoplasmic antibody-associated vasculitis (AAV) and systemic lupus erythematosus (SLE), glomerulonephritis is a severe kidney complication driven by immune cells, including T cells. However, the mechanisms underlying T cell activation in these contexts remain elusive. Here we report that in patients with AAV and SLE, type I interferon (IFN-I) induces T cell differentiation into interferon-stimulated genes-expressing T (ISG-T) cells, which are characterized by an elevated IFN-I signature, an immature phenotype, and cytotoxicity in inflamed tissue. Mechanistically, IFN-I stimulates the expression of interferon regulatory factor 7 (IRF7) in T cells, which in turn induces granzyme B production. In mice, blocking IFN-I signaling reduces IRF7 and granzyme B expression in T cells, thus ameliorating glomerulonephritis. In parallel, spatial transcriptomic analyses of kidney biopsies from patients with AAV or SLE reveal an elevated ISG signature and the presence of ISG-T cells in close proximity to plasmacytoid dendritic cells, the primary producers of IFN-I. Our results from both patients and animal models thus suggest that IFN-I production in inflamed tissue may drive ISG-T cell differentiation to expand the pool of cytotoxic T cells in autoimmune diseases.

Patients with systemic autoimmune diseases, such as anti-neutrophil cytoplasmic antibody (ANCA)-associated vasculitis (AAV) and systemic lupus erythematosus (SLE), frequently develop immune-mediated kidney diseases, including ANCA-associated glomerulonephritis (ANCA-GN) and lupus nephritis (LN), respectively[1]. In these conditions, aberrant immune responses lead to proliferative glomerulonephritis, including crescentic glomerulonephritis (cGN). cGN is the most severe and rapidly progressive form of glomerulonephritis characterized by glomerular crescent formation and necrosis, and is a primary contributor to end-stage kidney disease, notably in ANCA-GN[1–5]. In cGN, T cells and myeloid cells are the dominant cellular infiltrates in the kidney and are thought to play key roles in disease progression through cytotoxicity and the production of cytokines and other effector molecules[2,6–8].

T cells are central to adaptive immunity and are associated with autoimmune disease development[9–11]. They can directly kill target cells

[1]III. Department of Medicine, University Medical Center Hamburg-Eppendorf, Hamburg, Germany. [2]Hamburg Center for Translational Immunology, University Medical Center Hamburg-Eppendorf, Hamburg, Germany. [3]Institute of Medical Systems Biology, Center for Biomedical AI (bAIome), Center for Molecular Neurobiology (ZMNH), University Medical Center Hamburg-Eppendorf, Hamburg, Germany. [4]Haburg Center for Kidney Health, University Medical Center Hamburg-Eppendorf, Hamburg, Germany. [5]Institute for Immunology, University Medical Center Hamburg-Eppendorf, Hamburg, Germany. ✉e-mail: panzer@uke.uni-hamburg.de; n.asada@uke.de

through cytotoxic activity and orchestrate immune responses through cytokine production. Recent studies have highlighted the significant involvement of cytotoxic T lymphocytes (CTLs) in glomerulonephritis. CD8[+] T cells are associated with disease progression in ANCA-GN[12,13] and LN[14]. Moreover, clonal expansion of CD8[+] T cells is reported in both diseases[15,16], indicating CD8[+] T cells are activated in an antigen-specific manner. Furthermore, a murine cGN model study revealed the pathogenic role of the cytotoxic molecule, granzyme B (GzmB)[15]. However, the mechanisms driving T cell activation and the subsequent development of CTLs have remained elusive, posing a challenge in developing a therapeutic approach targeting cytotoxic T cells.

In this study, using single-cell RNA sequencing (scRNA-seq) and spatial transcriptomics, we identify interferon-stimulated gene-expressing T cells (ISG-T cells), which exhibit cytotoxicity in inflamed kidney tissue and reside in close proximity to plasmacytoid dendritic cells. Our findings identify ISG-T cells as a novel T cell subset that links local type I interferon (IFN-I) production to T cell differentiation during tissue inflammation.

## Results

### Identification of interferon-stimulated genes-expressing T cells in SLE and AAV patients

To explore T cell activation and transcriptomic changes in systemic autoimmune diseases, we analyzed our in-house and published scRNA-seq data of CD3[+] blood T cells from patients with AAV and SLE, as well as from healthy donors[17–20] (Fig. 1a). Using unsupervised clustering on these datasets, we identified a T cell cluster characterized by the expression of the type I interferon-stimulated genes (ISGs)[21], such as *ISG15*, *IRF7*, and *IFIT1* (Fig. 1b–d and Supplementary Fig. 1a–f). This ISG-expressing T (ISG-T) cell cluster encompassed both CD4[+] and CD8[+] T cells (Supplementary Fig. 1g, h). Enrichment analysis based on differentially expressed genes revealed an upregulation of IFN-I-associated pathways in ISG-T cells (Fig. 1e). The ISG-T cell population expanded in the blood of patients compared to healthy donors (mean: 0.19% vs 0.32% in ANCA dataset, 0.89% vs 3.61% in SLE dataset) (Fig. 1f and Supplementary Fig. 1i, j). In SLE, which is a condition associated with high levels of IFN-I signaling[22,23], we identified another cluster expressing ISGs and cytotoxicity-associated genes (Fig. 1b–d). This cluster, annotated as ISG-CTL, also slightly expanded in the blood of SLE patients compared to healthy donors (mean: 0.33% vs 0.46%) (Fig. 1f and Supplementary Fig. 1j). The expression of ISGs in ISG-T cell and ISG-CTL clusters indicated that these T cells were activated by IFN-I. To confirm the assumption that T cell activation by IFN-I leads to ISG expression, we analyzed publicly available transcriptome datasets of human and murine T cells stimulated with IFN-I[24,25]. In both human and murine T cells, stimulation with IFN-I resulted in the expression of ISGs, including *ISG15*, *ISG20*, *IFIT1*, and *IRF7* (Supplementary Fig. 2a, b). These findings suggest that T cells activated by IFN-I differentiate into the ISG-T/ISG-CTL cells, characterized by a distinct transcriptome signature including ISG expression.

### ISG-T cells exhibit an immature surface marker phenotype

To better characterize ISG-T cells, we investigated surface protein expression by analyzing our single-cell Cellular Indexing of Transcriptomes and Epitopes by Sequencing (scCITE-seq) data of T cells from AAV patients[15]. Using barcoded antibodies, this approach enabled us to evaluate mRNA expression and surface protein levels at single-cell resolution. Following unbiased clustering and annotating ISG-T cells based on their transcriptomic profiles, we analyzed the surface protein expression. ISG-T cells were CD45RA[-], CD45RO[±], CD44[+], CD62L[+], CD69[-], and CD279/PD1[-] (Supplementary Fig. 3a–c). In addition, we investigated blood ISG-T cells from treatment-naïve SLE patients (n = 3) and healthy donors using flow cytometry. We used IRF7 as a marker of ISG-T cells since *IRF7* mRNA, a well-characterized ISG, was consistently expressed in the ISG-T cell cluster across

multiple datasets (Fig. 1c and Supplementary Figs. 1c, d; 2a, b; 3c). While IRF7 expression was not observed in T cells from healthy donors (Supplementary Fig. 3d), we identified IRF7-expressing ISG-T cells in both CD4[+] and CD8[+] T cells in the blood of SLE patients (Supplementary Figs. 3d; 4). These ISG-T cells were characterized as CD45RA[sl+], CD45RO[-], CD44[-/±], CCR7[+], CD69[-], and CD279/PD1[-] (Supplementary Fig. 3e–g). These findings suggest that ISG-T cells exhibit an immature T cell phenotype similar to naïve T cells and central memory T cells, with slight variations between diseases and patients.

### ISG-T cells exhibit a cytotoxic phenotype in kidney tissue in murine glomerulonephritis models

In SLE and AAV, kidney involvement is a common finding, significantly impacting patient prognosis. To elucidate the role of ISG-T cells in immune-mediated kidney diseases, we used a well-established murine cGN model[18,19] (Fig. 2a). Ten days following cGN induction, we observed glomerular crescents and infiltration of CD4[+] and CD8[+] lymphocytes (Fig. 2b, c). Real-time RT-PCR analysis of kidney tissue revealed a significant upregulation of ISGs at days 10 and 20 (Supplementary Fig. 5a), indicating increased IFN-I signaling during cGN. To investigate T cell populations and their transcriptome at single-cell resolution, we isolated CD3[+] T cells from nephritic (cGN at day 10) and healthy kidneys, and performed scRNA-seq analysis. Mirroring our findings in patients with AAV and SLE in Fig. 1, unsupervised clustering of kidney T cells identified ISG-T cells (Fig. 2d–g). These kidney ISG-T cells, comprising both CD4[+] and CD8[+] T cells, exhibited high levels of ISGs, including *Isg15*, *Irf7*, and *Ifit1* (Fig. 2e–g and Supplementary Fig. 5b–d). Notably, kidney ISG-T cells expressed cytotoxic genes including *Gzmb* and displayed an elevated cytotoxicity score, positively correlating with the ISG score within the cluster (Fig. 2e–g and Supplementary Fig. 5e–h). Enrichment analysis based on differentially expressed genes showed significant upregulation of IFN-I-associated pathways in ISG-T cells (Fig. 2h). Compared to healthy controls, ISG-T cells and other T cell populations, including CD8[+] Trm cells, CD4[+] Th17 cells, Th2/Treg cells, and γδ T cells, expanded in nephritic conditions (Fig. 2i). In the cGN group, ISG-T cells exhibited a higher ISG score compared to controls (Fig. 2j and Supplementary Fig. 5i), which is in line with the elevated IFN-I signature in the kidney during cGN (Supplementary Fig. 5a). In addition, to investigate kidney T cells in a lupus nephritis model, we analyzed a published scRNA-seq dataset of T cells in MRL/lpr lupus-prone nephritic mice[26] (Supplementary Fig. 6a). As observed in the cGN model, unsupervised clustering revealed an ISG-T cell cluster (Supplementary Fig. 6b), with high ISG and cytotoxicity scores (Supplementary Fig. 6c–h). Enrichment analysis showed upregulated IFN-I pathways (Supplementary Fig. 6i). In the kidney, ISG-T cells and CD8[+] Trm cells expanded compared to the spleen (ISG-T cells, 0.86% in the spleen and 1.12% in the kidney) (Supplementary Fig. 6j), indicating ISG-T cell activation in murine lupus nephritis. These findings demonstrate that ISG-T cells expand and adopt a cytotoxic phenotype in the kidney during immune-mediated kidney diseases.

### ISG-T cells show a low pseudotime score, indicating immaturity

We further characterized the CD8[+] ISG-T cell population using the scRNA-seq data of the murine cGN model. Reclustering of CD8[+] T cells revealed a distinct ISG-T cell population, characterized by high ISG and cytotoxicity scores (Fig. 3a, b and Supplementary Fig. 7a). To assess cell differentiation, we performed trajectory analysis using Monocle 3[27] by setting the naïve T cell cluster as a starting point of differentiation. The trajectory analysis revealed that the pseudotime score of ISG-T cells was as low as that of naïve T cells, whereas other clusters, such as CTL and Tem clusters, exhibited higher pseudotime scores (Fig. 3c). These data suggest that ISG-T cells are still immature, as already indicated by the surface marker analysis of ISG-T cells from patients with AAV or SLE (Supplementary Fig. 3a–g), and that these cells might further differentiate into more mature cell types. Next, we analyzed

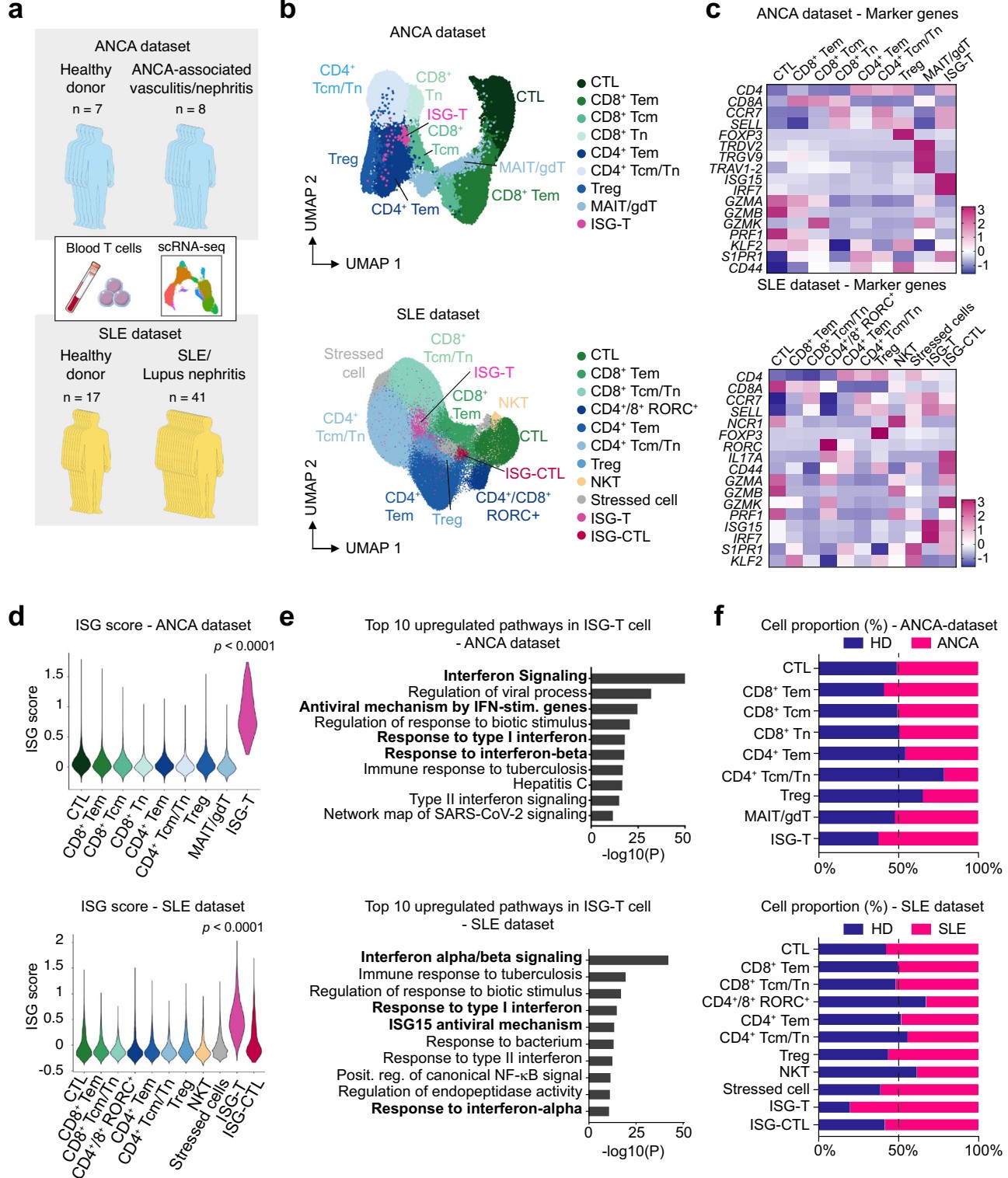

**Fig. 1 | ISG-T cells are expanded in the blood of patients with AAV and SLE.**
**a** Schematic overview of the experiment. scRNA-seq datasets of blood T cells from AAV or SLE patients and healthy donors (HD) were analyzed. **b** UMAP plots of blood T cells in the ANCA and SLE datasets. **c** Heatmaps showing the expression of marker genes (comparison between different T cell clusters). **d** Violin plots depicting the ISG scores in each cluster. The ISG score was calculated based on genes listed in Supplementary Fig. 1e, f. P values were calculated by the two-sided Wilcoxon rank-sum test. **e** Enrichment analysis showing the top 10 upregulated pathways in the ISG-T cell cluster. Gene ontology and pathway enrichment were analyzed using

Metascape. P-values were calculated using the cumulative hypergeometric test, and adjusted for multiple comparisons using the Benjamini-Hochberg method. **f** Graphs showing the proportions of healthy donors and patients in each cluster. Cell type abbreviations: ISG-T interferon-stimulated genes-expressing T, CTL cytotoxic lymphocytes, Tem effector memory T, Tcm central memory T, Tn naïve T, Treg regulatory T, MAIT mucosal-associated invariant T, gdT γδ T, NKT natural killer T, ISG-CTL interferon-stimulated genes-expressing cytotoxic lymphocyte. Source data are provided as a Source Data file.

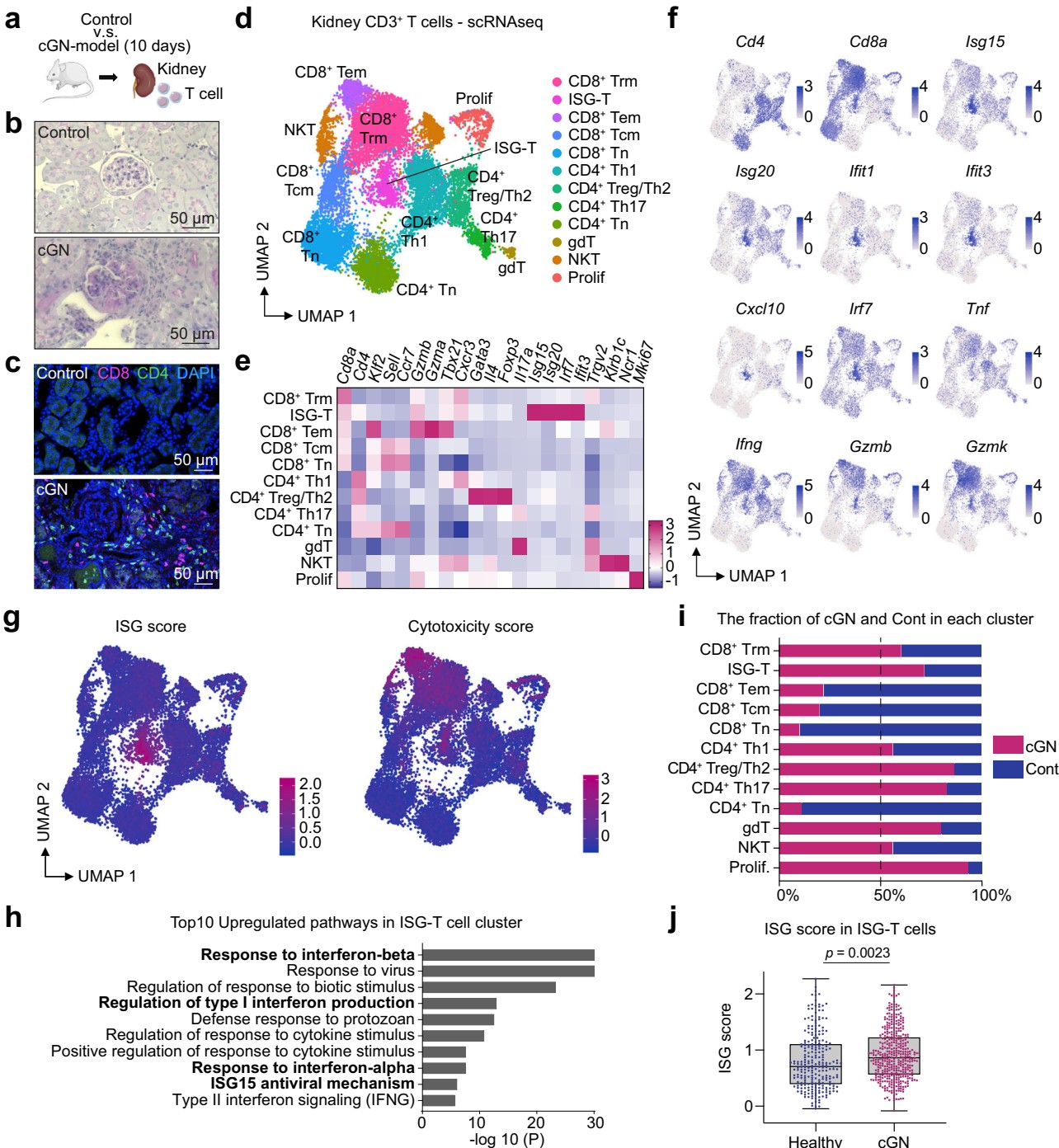

**Fig. 2 | Kidney ISG-T cells display cytotoxicity in a murine cGN model.**
**a** Schematic overview of the experiment. A cGN model was induced in mice, and kidneys from these mice were analyzed together with those from healthy controls. Representative images of PAS-stained kidney sections from the control and the cGN group (**b**) and of immunofluorescence staining showing CD4⁺ and CD8⁺ T cells in the kidney (**c**). These findings were confirmed three times. **d** UMAP plot of CD3⁺ T cells isolated from nephritic and healthy control kidneys. Heatmap (**e**) and UMAP plots (**f**) showing the marker gene expression for each cluster. **g** UMAP plots depicting the ISG and cytotoxicity scores. The ISG and cytotoxicity scores were calculated based on genes listed in Supplementary Fig. 5c, f, respectively. **h** Enrichment analysis illustrating the top 10 upregulated pathways in the ISG-T cell cluster. Gene ontology and pathway enrichment were analyzed using Metascape.

P-values were calculated using the cumulative hypergeometric test, and adjusted for multiple comparisons using the Benjamini-Hochberg method. **i** Bar graph showing the proportion of the cGN and control groups in each cluster. **j** Box plot showing the ISG score in ISG-T cells in the cGN and control groups (Healthy, n = 246 cells; cGN, n = 404 cells). Box plots show the median (horizontal line inside the box), the 25th and 75th percentiles (lower and upper bounds of the box), and the minimum and maximum values (whiskers). P value was calculated by unpaired two-tailed t-test with Welch's correction. Cell type abbreviations: ISG-T interferon-stimulated genes-expressing T, Trm tissue resident memory T, Tem effector memory T, Tcm central memory T, Tn naïve T, Treg regulatory T, gdT γδ T, NKT natural killer T, Prolif proliferating cells. Source data are provided as a Source Data file.

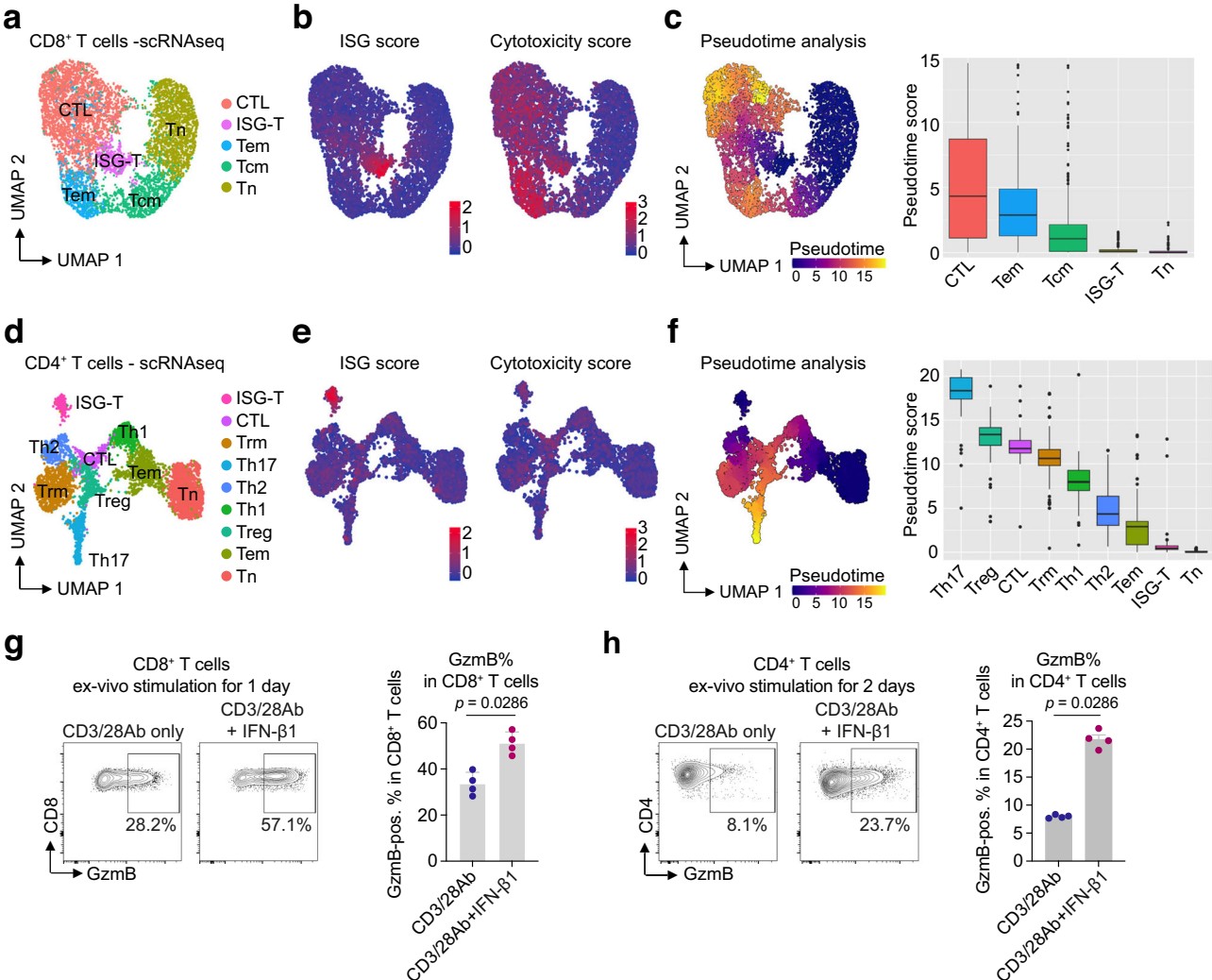

**Fig. 3 | IFN-I promotes T cell differentiation into cytotoxic T cells. a** UMAP plot of CD8+ T cells from nephritic and healthy kidneys. **b** UMAP plots showing ISG and cytotoxicity scores in CD8+ T cells. **c** Pseudotime analysis using Monocle 3. Naïve CD8+ T cell cluster was set to the starting point of differentiation (root of the pseudotime). Quantification is also shown (CTL, n = 2985; Tem, n = 746; Tcm, n = 1300; ISG-T, n = 437; Tn, n = 1855). **d** UMAP plot of CD4+ T cells from nephritic and healthy kidneys. **e** UMAP plots depicting ISG and cytotoxicity scores in CD4+ T cells. **f** Pseudotime analysis using Monocle 3. Naïve CD4+ T cell cluster was set to the starting point of differentiation. Quantification is also shown (Th17, n = 381; Treg, n = 433; CTL, n = 248; Trm, n = 977; Th1, n = 473; Th2, n = 332; Tem, n = 798; ISG-T, n = 210; Tn, n = 1331). Representative flow cytometry plots and bar graphs

showing the production of GzmB in CD8+ T cells (**g**) and in CD4+ T cells (**h**) stimulated with vehicle or IFN-β1 in combination with CD3/28 Ab (n = 4 mice). **g**, **h** P values were calculated by the two-sided Mann-Whitney test. Data are mean + S.E.M. Box plots show the median (horizontal line inside the box), the 25th and 75th percentiles (lower and upper bounds of the box), and the whiskers extend to the most extreme data points not considered outliers (within 1.5 × IQR). Outliers beyond this range are shown as individual dots. Cell type abbreviations: ISG-T interferon-stimulated genes-expressing T, CTL cytotoxic lymphocytes, Trm tissue-resident memory T, Tem effector memory T, Tcm central memory T, Tn naïve T, Treg regulatory T. Source data are provided as a Source Data file.

CD4+ T cells. Similar to the findings in CD8+ T cells, CD4+ T cell reclustering identified an ISG-T cell population and cytotoxic T cells (Fig. 3d and Supplementary Fig. 7b). The ISG-T cell cluster exhibited a high ISG score, and both ISG-T and CTL clusters showed moderately elevated cytotoxicity scores (Fig. 3e and Supplementary Fig. 7b). As observed in CD8+ T cells, CD4+ ISG-T cell cluster also exhibited a low pseudotime score, indicating immaturity and the potential for differentiation into more mature populations (Fig. 3f).

**IFN-I induces the expression of GzmB in both CD8+ and CD4+ T cells**

The expression of cytotoxic genes in kidney ISG-T cells suggests that IFN-I induces T cell cytotoxicity. To test this hypothesis, we stimulated CD8+ T cells ex vivo with anti-CD3/28 antibodies in combination with either vehicle or IFN-β1. Stimulation with IFN-β1 for 1 day significantly

enhanced GzmB production in CD8+ T cells compared to vehicle (Fig. 3g). While GzmB production in CD4+ T cells remained negligible after 1 day of stimulation with IFN-β1 (Supplementary Fig. 7c), ~20% of CD4+ T cells produced GzmB after 2 days of stimulation with IFN-β1 (Fig. 3h), indicating that IFN-I also plays a role in promoting a cytotoxic phenotype in CD4+ T cells. Moreover, surface CD107a expression, a marker of T cell degranulation, also significantly increased in IFN-I-treated CD8+ and CD4+ T cells, indicating enhanced cytotoxic function (Supplementary Fig. 7d). These findings were not observed in *Ifnar1* knockout T cells, which lack the gene encoding IFNAR1, the receptor for IFN-I[28,29] (Supplementary Fig. 7e, f). We also examined the effect of IL-12, which is known to be crucial to CD8+ T cell differentiation[24]. In contrast to IFN-I, IL-12 did not induce GzmB production in CD8+ T cells. However, IL-12 significantly increased IFN-γ production in CD8+ T cells compared to vehicle or IFN-β1 groups (Supplementary Fig. 7g, h),

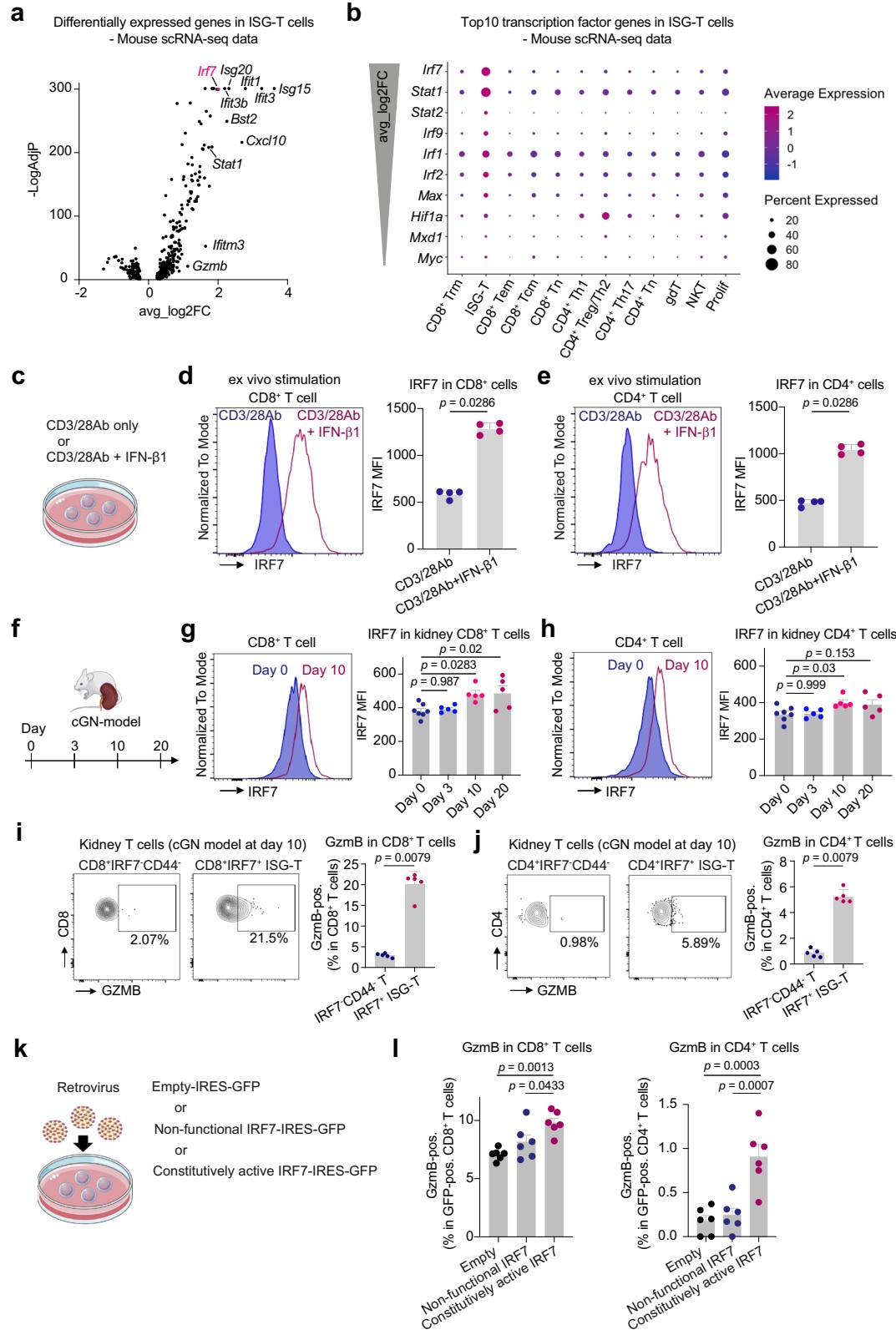

indicating distinct roles for these cytokines in CD8⁺ T cell differentiation. Collectively, these findings underscore that IFN-I drives T cell cytotoxicity.

## IRF7 promotes T cell differentiation towards cytotoxic T cells

Next, we investigated the molecular mechanism driving T cell cytotoxicity in response to IFN-I. Among significantly upregulated genes in

the ISG-T cell cluster, *Irf7* was identified as the transcription factor with the highest expression levels (Fig. 4a, b). IRF7 is a member of interferon regulatory factors (IRFs)[30] and plays a key role in the production of IFN-I in myeloid cells and fibroblasts[31]. While the role of IRF7 in CD8⁺ T cell expansion is documented in a viral infection model[32], its importance in T cell cytotoxicity and immune mediated diseases is not well understood. Therefore, we characterized IRF7 expression in T cells in in-vitro

**Fig. 4 | IRF7 is associated with cytotoxicity in T cells stimulated with IFN-I.**
**a** Volcano plot illustrating the significantly upregulated and downregulated genes
in ISG-T cells. **b** Dot plot showing the top 10 transcription factor genes expressed in
ISG-T cells. **c** T cells were stimulated with vehicle or IFN-β1 in combination with
CD3/28 Ab. Representative histograms showing the expression of IRF7 in CD8⁺
T cells (**d**) and CD4⁺ T cells (**e**). Quantification of IRF7 MFI is also shown (n = 4 mice).
**f** cGN was induced in mice, and kidney T cells were analyzed on days 0, 3, 10, and
20. Representative histograms depicting the expression of IRF7 in CD8⁺ T cells (**g**)
and CD4⁺ T cells (**h**) at cGN day 0 and day 10. Quantification of IRF7 MFI at each
timepoint is also shown (Day 0, n = 7; Days 3, 10, and 20, n = 5). IRF7⁺ and IRF7⁻CD44⁻

kidney T cells were analyzed for the production of GzmB in CD8⁺ T cells (**i**) and CD4⁺
T cells (**j**) at cGN day 10 (n = 5 mice). **k** T cells were retrovirally transduced with
constitutively active IRF7-GFP, non-functional IRF7-GFP, or GFP only. **l** Bar graphs
showing the quantification of GzmB-positive cells in CD8⁺ and CD4⁺ T cells (n = 6
mice). **a** Differentially expressed genes were identified using Seurat's FindMarkers
function with the Wilcoxon rank sum test. P-values were adjusted for multiple
testing using the Benjamini-Hochberg method. **g, h, l** P values were calculated using
one-way ANOVA with Tukey's multiple comparison test. **d, e, i, j** P values were
calculated by the two-sided Mann-Whitney test. Data are mean + S.E.M. Source data
are provided as a Source Data file.

experiments using IFN-β1 and in the cGN model. In in-vitro experiments, IRF7 expression was strongly induced by IFN-β1 in both CD8⁺ and CD4⁺ T cells (Fig. 4c–e), which was consistent with our analysis of transcriptome data of T cells stimulated with IFN-α (Supplementary Fig. 2). This upregulation of IRF7 expression by IFN-β1 was not observed in *Ifnar1* KO T cells (Supplementary Fig. 7i). In experiments using the murine cGN model, flow cytometry analysis of kidney T cells revealed that IRF7 was highly expressed in CD8⁺ and CD4⁺ T cells 10 days after cGN induction (Fig. 4f–h). The frequency of GzmB-producing T cells was significantly higher in IRF7⁺ ISG-T cells compared to IRF7⁻CD44⁻ T cells at day 10 post-cGN induction (Fig. 4i, j). To investigate the effect of IRF7 on T cell cytotoxicity, we transduced T cells with constitutively active IRF7-IRES-GFP, non-functional IRF7-IRES-GFP, or GFP alone using a retrovirus system[33,34] (Fig. 4k). Transduction of T cells with constitutively active IRF7 increased GzmB expression in both CD8⁺ T cells (mean: 7.1% in GFP alone vs 9.8% in constitutively active IRF7, $p < 0.001$) and CD4⁺ T cells (mean: 0.18% in GFP alone vs 0.9% in constitutively active IRF7, $p < 0.001$) compared to controls (Fig. 4l). Following the transduction of T cells with constitutively active IRF7, the expression of *Stat1*, a key transcription factor in T cell differentiation[30], significantly increased (Supplementary Fig. 7j). These data suggest that IFN-I drives T cell cytotoxicity, at least in part, through the action of IRF7.

## Blocking IFN-I signaling reduces the expression of IRF7 and GzmB in T cells in vivo and attenuates cGN

To investigate the role of IFN-I signaling in T cell activation in vivo, we performed adoptive T cell transfer experiments from either wild-type (WT) or *Ifnar1* KO mice into *Rag1* KO mice, which lack lymphocytes. Two weeks after the T cell reconstitution, a cGN model was induced, and mice were analyzed 10 days after disease induction (Fig. 5a). In the *Ifnar1* KO T cell-transferred group, mice developed less severe glomerulonephritis compared to WT T cell-transferred group (Fig. 5b and Supplementary Fig. 8a, b). In addition, *Ifnar1* KO T cells exhibited reduced levels of IRF7 expression and cytotoxicity (Fig. 5c–f and Supplementary Fig. 8c, d). These data indicate that blocking IFN-I signaling specifically in T cells reduces their cytotoxicity in vivo and leads to an ameliorated course of cGN. Next, we investigated the therapeutic potential of targeting IFN-I signaling in the cGN model by blocking IFNAR1 using neutralizing antibodies. Following the induction of cGN, we intraperitoneally injected anti-IFNAR1 antibodies (Fig. 5g). The antibody treatment significantly attenuated the formation of glomerular crescents (Fig. 5h). Although albuminuria and BUN levels also decreased, these changes did not reach statistical significance (Supplementary Fig. 8e, f). Moreover, IFNAR1 neutralization significantly reduced IRF7 levels in CD8⁺ and CD4⁺ T cells (Fig. 5i, j), decreased the frequency of GzmB-positive cells in CD8⁺ T cells (mean: 6.6% vs 3.4%, $p < 0.01$) and CD4⁺ T cells (mean: 1.8% vs 1.3%, $p = 0.10$) (Fig. 5k, l), and diminished the infiltration of immune cells (Supplementary Fig. 8g–j). Altogether, these findings indicate that IFN-I signaling induces T cell cytotoxicity in vivo and that targeting this pathway could offer therapeutic benefits in cGN.

## Inflamed kidney tissue as a potential site for T cell activation by IFN-I

To determine the mode of T cells activation by IFN-I during cGN, we performed scRNA-seq analyses on CD45⁺ leukocytes from the kidney, spleen, and blood, as well as CD45⁻ kidney cells, isolated from nephritic mice 10 days after cGN induction (Fig. 6a–d and Supplementary Fig. 9a–d). None of these datasets showed detectable *Ifna1* expression. While *Ifnb1* expression was not seen in the datasets of spleen CD45⁺ cells, blood CD45⁺ cells, or kidney CD45⁻ cells, it was detected in a kidney macrophage/dendritic cell cluster in the dataset of kidney CD45⁺ cells (Fig. 6a–d). The expression of IFN-I receptor genes, *Ifnar1* and *Ifnar2*, was observed in most leukocytes, including T cells (Supplementary Fig. 9e). Given that *Ifnb1* expression was detected exclusively in the kidney macrophage/dendritic cell cluster, we hypothesized that T cell activation by IFN-I takes place in the inflamed kidney. The IFN-I signaling from the macrophage/dendritic cell cluster to T cells was indicated by a cell-cell interaction analysis using CellChat[35] (Supplementary Fig. 9f). To confirm that T cells are activated by IFN-I in inflamed kidney tissues, we analyzed IRF7 levels in T cells across organs using flow cytometry. IRF7 levels in CD8⁺ and CD4⁺ T cells were significantly higher in the kidney compared to blood and spleen 10 days after cGN induction (Fig. 6e). Correspondingly, GzmB production in T cells increased only in the kidney following cGN induction (Fig. 6f). These findings suggest that local IFN-I signaling in the kidney may play an important role in T cell activation during immune-mediated kidney disease.

## IFN-I signaling and cytotoxicity are associated with tissue inflammation in immune-mediated kidney diseases

To investigate the clinical relevance of our findings derived from murine experiments, we analyzed kidneys from patients with immune-mediated kidney diseases. First, we examined transcriptome data of kidney biopsies using the European Renal cDNA Bank (ERCB) database[36] (Fig. 7a). Compared with healthy kidney tissue, ISGs were highly expressed in the kidney tissues from patients with ANCA-GN and lupus nephritis (Fig. 7b), indicating increased IFN-I signaling in both diseases. Similarly, genes coding for cytotoxic molecules were highly expressed in kidneys from both ANCA-GN and lupus nephritis groups (Fig. 7b). Of particular note, the ISG score positively correlated with the cytotoxicity score in both ANCA-GN and lupus nephritis patients (Fig. 7c), suggesting that kidney tissues with higher IFN-I signaling harbor more cytotoxic lymphocytes. In addition, a kidney injury marker *SPP1*[37,38], which codes for Osteopontin, showed a positive correlation with the ISG score and cytotoxicity score (Fig. 7d, e). To further investigate T cell activation in inflamed kidney tissue, we isolated T cells from the kidney and matched blood samples of 27 ANCA-GN patients, and performed scRNA-seq analyses (Fig. 8a). ISG-T cells were identified by their high ISG expression levels (Fig. 8b, c and Supplementary Fig. 10a, b). Notably, the ISG-T cells in the nephritic kidney exhibited significantly higher ISG and cytotoxicity scores compared to those in the blood (Fig. 8d), suggesting that ISG-T cells in the kidney are more strongly activated by IFN-I than those in the blood in ANCA-GN patients.

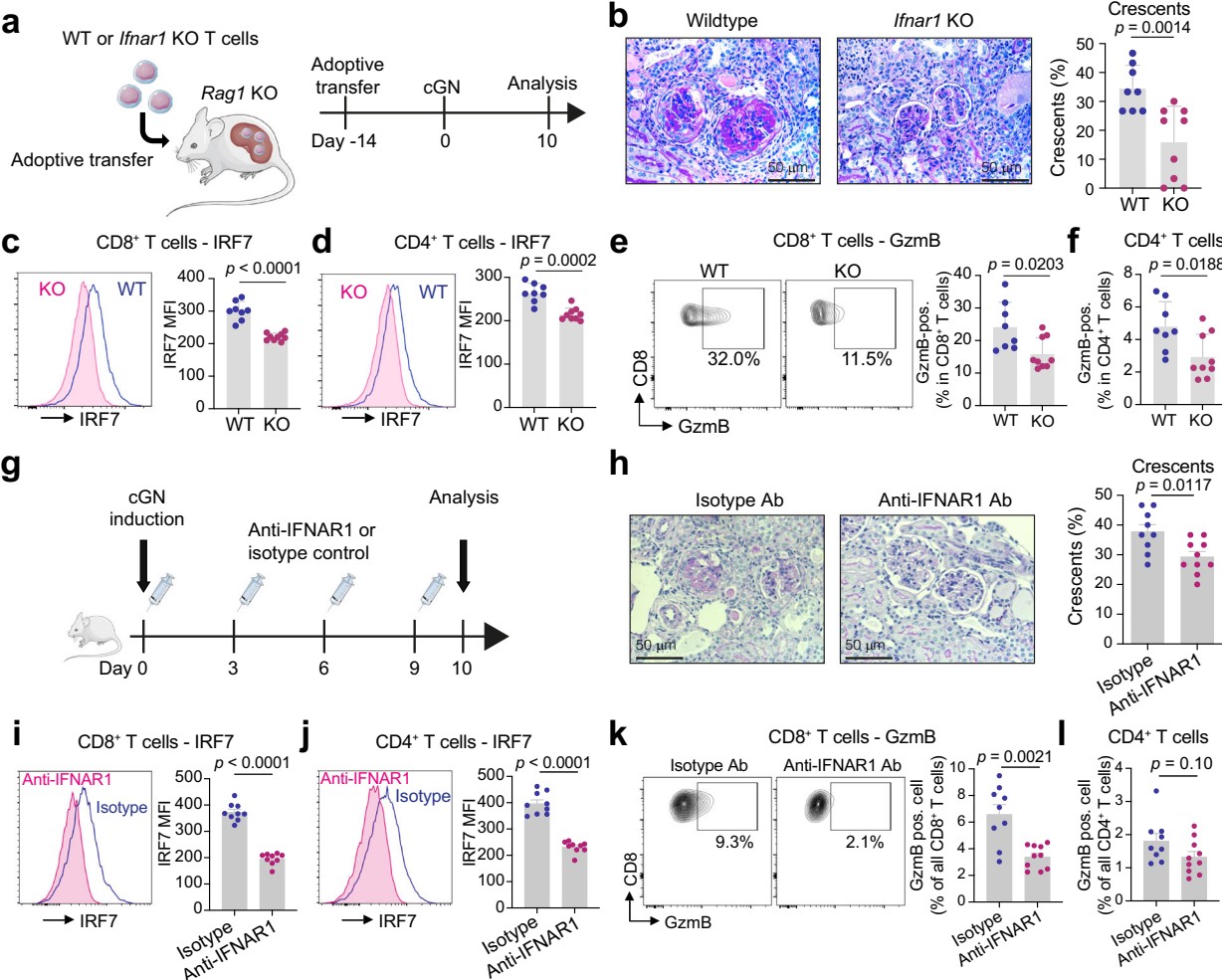

**Fig. 5 | IFN-I signaling in T cells drives GzmB production and renal tissue damage. a** CD8+ and CD4+ T cells were isolated from the spleen of *Ifnar1* KO or wild-type mice, and transferred into *Rag1* KO mice (WT, n = 8; KO, n = 9). **b** Representative photographs of PAS staining of the kidney. Quantification is also shown (WT, n = 8; KO, n = 9). Representative histograms and bar graphs showing the MFI of IRF7 in CD8+ T cells (**c**) and CD4+ T cells (**d**) isolated from the kidney (WT, n = 8; KO, n = 9). **e** Representative flow cytometry plots and a bar graph illustrating the production of GzmB in CD8+ T cells (WT, n = 8; KO, n = 9). **f** Bar graph showing the production of GzmB in CD4+ T cells (WT, n = 8; KO, n = 9). **g** cGN was induced in mice, and mice were injected with anti-IFNAR1 Ab (n = 10) or isotype control (n = 9)

every 3 days. **h** Representative photographs of PAS-stained kidney sections from the isotype control and anti-IFNAR1 Ab groups. Quantification of glomerular crescents is also shown (anti-IFNAR1 Ab, n = 10; isotype Ab, n = 9). Representative histograms and bar graphs showing the MFI of IRF7 in CD8+ T cells (**i**) and CD4+ T cells (**j**) isolated from the nephritic kidneys (anti-IFNAR1 Ab, n = 10; isotype Ab, n = 9). **k** Representative flow cytometry plots and a bar graph showing the GzmB production in CD8+ T cells (anti-IFNAR1 Ab, n = 10; isotype Ab, n = 9). **l** Bar graph showing the GzmB production in CD4+ T cells (anti-IFNAR1 Ab, n = 10; isotype Ab, n = 9). Data are mean + S.E.M. P values were calculated by the two-sided Mann-Whitney test. Source data are provided as a Source Data file.

Next, to investigate the relationship between IFN-I signaling and tissue inflammation in different tissue compartments, we examined the kidney biopsies from 28 ANCA-GN patients using sequencing-based spatial transcriptomics (Fig. 8e). We identified inflamed tubulointerstitial and glomerular areas in addition to healthy compartments (Fig. 8f, g and Supplementary Fig. 10c). Inflamed tubulointerstitial and glomerular areas exhibited significantly higher *IRF7* expression, ISG and cytotoxicity scores, and estimated T cell abundance than healthy areas (Fig. 8f–j and Supplementary Fig. 10c–f). These findings indicate that tissue inflammation is associated with IFN-I signaling and cytotoxicity in ANCA-GN.

## ISG-T cells exist in close proximity to plasmacytoid dendritic cells in the kidney of ANCA-GN and lupus nephritis

Finally, to investigate the presence and localization of ISG-T cells in inflamed kidney tissue at single-cell resolution, we employed an imaging-based single-cell spatial transcriptomics technique[39]. This analysis included kidney tissues from 32 ANCA-GN patients, 19 lupus

nephritis patients, and 6 controls (Fig. 9a). For the control group, non-tumor kidney tissues obtained from tumor nephrectomy specimens were used. Unsupervised clustering identified 40 different cell types (Fig. 9b–e and Supplementary Fig. 11a, b). Among these, ISG-T cells were identified by their significantly higher levels of ISGs compared to other cell types (Fig. 9b–f and Supplementary Fig. 11c). In both ANCA-GN and lupus nephritis, a subset of patients exhibited a higher percentage of ISG-T cells among T cells (Fig. 9g). To investigate cell-cell interactions in kidney tissues, we conducted cell proximity analysis by calculating the distances between different cell types. Notably, this analysis revealed that plasmacytoid dendritic cells (pDC), the primary producers of IFN-I[30,40], were the closest neighbors of ISG-T cells among non-lymphocyte cells in both ANCA-GN and lupus nephritis (Fig. 9h). The distance between ISG-T cells and pDCs was significantly smaller in ANCA-GN and lupus nephritis compared to controls (median distances: 98.85 μm in ANCA-GN, 67.11 μm in lupus nephritis, and 478.64 μm in controls) (Fig. 9i and Supplementary Fig. 11d). Among lymphocytes, naïve T cells were identified as the closest neighbors to

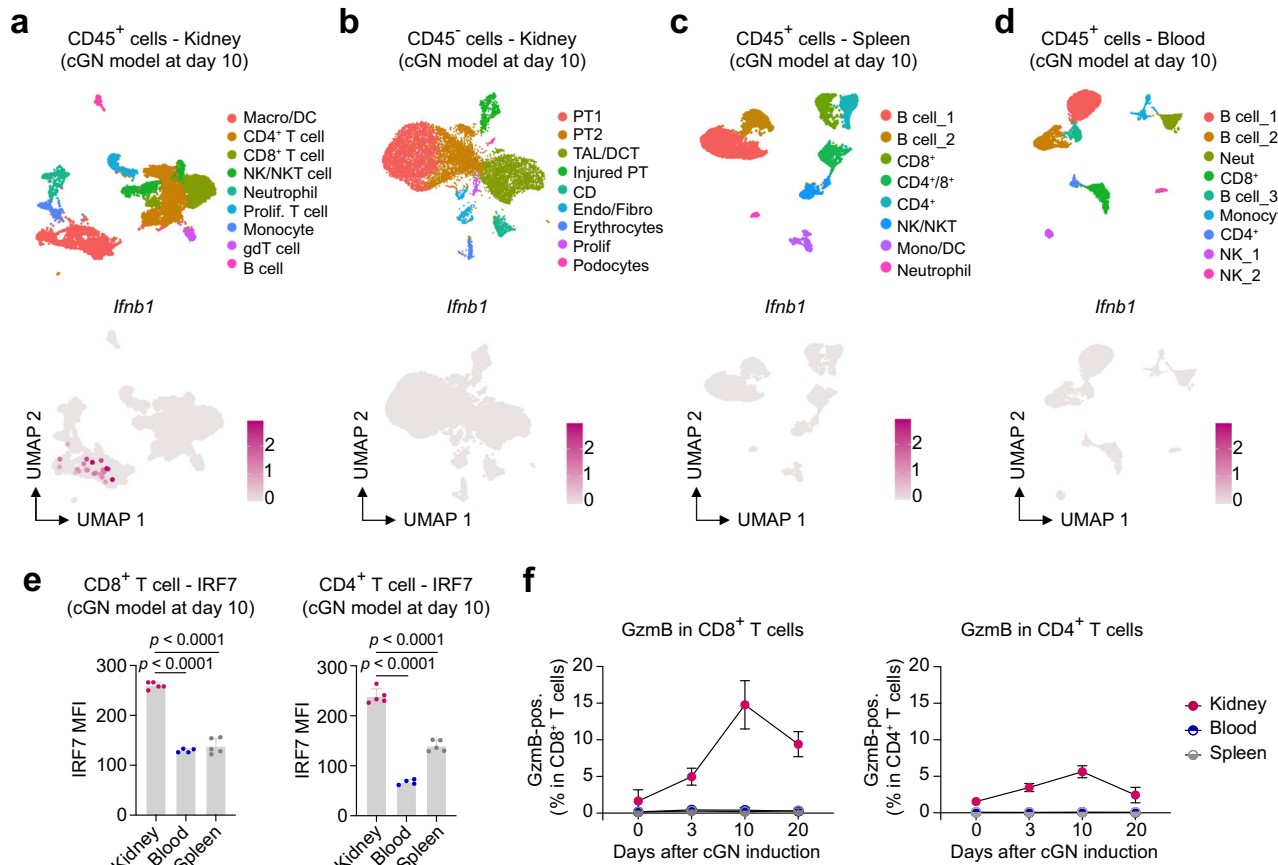

**Fig. 6 | T cells exhibit an IFN-I response in the nephritic kidney in the cGN model.** UMAP plots showing different cell types: CD45$^+$ cells (**a**) and CD45$^-$ cells (**b**) from the nephritic kidneys, and CD45$^+$ cells from the spleen (**c**) and blood (**d**). *Ifnb1* expression is also shown. Cells were analyzed 10 days after cGN induction. **e** IRF7 expression in CD8$^+$ and CD4$^+$ T cells analyzed with flow cytometry. T cells were isolated from the kidney, blood, and spleen 10 days after cGN induction (kidney and spleen, n = 5; blood, n = 4). P values were calculated using one-way

ANOVA with Tukey's multiple comparison test. Data are mean + S.E.M. **f** Time-course plot showing GzmB expression in CD8$^+$ and CD4$^+$ T cells isolated from the kidney, blood, and spleen on days 0, 3, 10, and 20 (days 0, 3, 20, n = 5; day 10, n = 4). Data are mean ± S.D. Cell type abbreviations: Macro macrophage, DC dendritic cell, NK natural killer, NKT natural killer T, gdT γδ T, PT proximal tubule, TAL thick ascending limb, DCT distal convoluted tubule, CD collecting duct, Endo endothelial cell, Fibro fibroblast. Source data are provided as a Source Data file.

ISG-T cells, potentially reflecting their differentiation from naïve T cells (Supplementary Fig. 11e). Collectively, these findings demonstrate the presence of ISG-T cells in close proximity to pDCs in ANCA-GN and lupus nephritis, supporting the possibility of local T cell activation by IFN-I in inflamed kidney tissues.

## Discussion

Our study provides novel insights into the role of IFN-I in T cell differentiation into ISG-T cells in autoimmune diseases. ISG-T cells showed an immature and cytotoxic phenotype, particularly in the nephritic kidneys. Furthermore, through analyses of kidney biopsies from patients with ANCA-GN or lupus nephritis and a murine cGN model, we revealed that ISG-T cells might be locally activated in inflamed kidney tissue by IFN-I-producing myeloid cells, such as pDCs.

The role of IFN-I is emerging as a shared feature in autoimmune diseases such as SLE, AAV, Sjögren syndrome, myositis, and systemic sclerosis[17,22,23,40,41]. Clinical trials showed the efficacy of anifrolumab, a human monoclonal antibody targeting the IFN-I receptor, in SLE patients[42–44]. Yet, the role of IFN-I in T cell activation has not been studied in detail[20,45]. Our analysis revealed an expansion of ISG-T cells in the blood of AAV and SLE patients, indicating that IFN-I has a unique role in T cell activation in these diseases.

IFN-I induced the expression of ISGs and cytotoxic molecules in T cells (Figs. 3g, h, 4d, e). In contrast, IL-12, a well-characterized cytokine for CD8$^+$ T cell differentiation[24], promoted the production of IFN-γ

but not GzmB (Supplementary Fig. 7g, h). Based on these findings, we hypothesize that IFN-I drives T cell differentiation into ISG-T cells to efficiently expand the pool of CTLs (but not IFN-γ-producing T cells) in diseases where IFN-I is produced. Recently, a study reported that IFN-I induces clonal expansion of CD38$^+$CD8$^+$ CTLs in immune checkpoint inhibitor (ICI)-induced arthritis[46]. In our analyses, *CD38* expression was detected only in the ISG-CTL cluster of the SLE dataset but not in the ISG-T cells (Supplementary Fig. 12a). Pseudotime analysis revealed that the ISG-CTL cluster exhibits a higher pseudotime score compared to ISG-T cells (Supplementary Fig. 12b), indicating that ISG-T cells may differentiate into CD38$^+$ ISG-CTLs. However, further studies are required to validate this hypothesis.

In certain autoimmune diseases, such as systemic sclerosis and IgG4-related disease, an expansion of CD4$^+$ CTLs was reported[47,48]. scRNA-seq analysis of blood lymphocytes from patients with IgG4-related disease revealed the presence of not only CD4$^+$ CTLs but also CD4$^+$ T cells expressing ISGs[48]. Given that IFN-I induced cytotoxicity in both CD8$^+$ and CD4$^+$ T cells in our in-vitro experiments, IFN-I likely plays a key role in CD4$^+$ T cell activation in diseases in which CD4$^+$ CTLs are expanded. Moreover, CTL differentiation through ISG-T cells may play a crucial role in viral infections, where IFN-I is essential for host defense. In a publicly available scRNA-seq dataset of patients with SARS-CoV-2[49], we identified an ISG-T cell cluster with ISG and cytotoxicity scores higher than those in healthy donors (Supplementary Fig. 13).

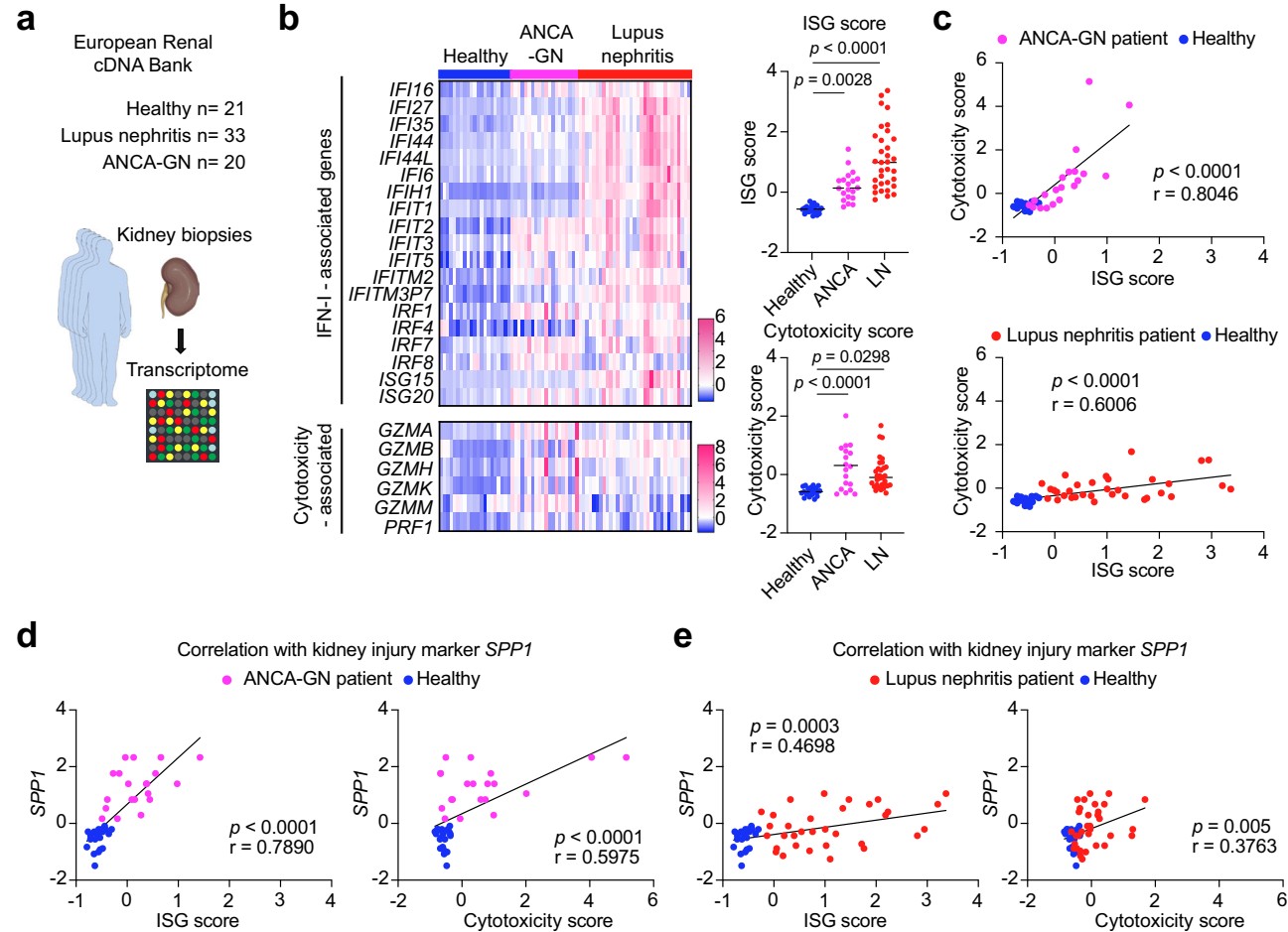

**Fig. 7 | ISG and cytotoxicity scores are increased in the kidney in ANCA-GN and lupus nephritis. a** Transcriptome data of kidney biopsies from healthy controls, lupus nephritis, and ANCA-GN were analyzed. **b** Heatmap showing the expression of IFN-I and cytotoxicity-associated genes in the kidney. Graphs show the quantification of ISG and cytotoxicity scores based on the genes in the heatmap (Control, n = 21; Lupus nephritis, n = 33; ANCA-GN, n = 20). **c** Scatter plots showing the correlation between ISG score and cytotoxicity score. Each dot represents a patient. Scatter plots showing the correlation of *SPP1* with ISG score or cytotoxicity score in ANCA-GN (**d**) and in lupus nephritis (**e**). **b** P values were calculated by one-way ANOVA with Tukey's multiple comparison test. **c**–**e** P values were calculated using Pearson correlation with a two-sided test. Source data are provided as a Source Data file.

ISG-T cells exhibited higher cytotoxicity in the kidney compared to the blood in patients with ANCA-GN, which might be attributable to elevated IFN-I levels in inflamed kidney tissue relative to those in the blood. However, unlike in SLE/lupus nephritis, where systemic IFN-I dysregulation is well-established, IFN-I signaling in AAV/ANCA-GN might be more localized to inflamed tissue. In our analysis, the expansion of ISG-T cells in the blood and the increased ISG score in kidney tissue were less pronounced in AAV/ANCA-GN patients compared to SLE/lupus nephritis patients (Figs. 1, and 7). These findings are consistent with a previous study that described the difficulty of detecting an IFN-I signature in peripheral blood mononuclear cells[50].

The degree of IFN-I-mediated inflammation may differ among patients within the same disease group. The lupus nephritis data in Figs. 7b and 9g showed that some SLE patients exhibited strikingly elevated ISG levels and ISG-T cell proportion in the kidney, likely reflecting the known heterogeneity within SLE patient groups. Additionally, in the spatial transcriptomics of ANCA-GN, a subset of patients showed a higher percentage of ISG-T cells among total T cells (Fig. 9g), suggesting that IFN-I-mediated inflammation occurs in a specific subtype of ANCA-GN. These findings indicate that patients may respond differently to therapies targeting IFN-I pathways. Further research is thus required to improve patient stratification and therapeutic approaches.

Myeloid cells, particularly pDCs, are reported to be the primary producers of IFN-I[30,40]. In our analysis of the murine cGN model, we did not observe a distinct pDC cluster, possibly due to the low number of pDCs. Nevertheless, IFN-I expression was exclusively detected in a cluster composed of macrophages and dendritic cells. In our imaging-based spatial transcriptomics analysis of ANCA-GN and lupus nephritis, pDCs were identified as the closest neighboring cell type to ISG-T cells among all non-lymphocyte cells, suggesting that ISG-T cells might be locally activated by pDCs in the kidney. Previous studies showed that other leukocytes and epithelial cells can also produce IFN-I in lupus nephritis and ANCA-GN[51,52]. Therefore, further analysis is needed to better characterize the cellular source of IFN-I in immune-mediated kidney diseases.

The important role of IFN-I in the progression of the murine cGN model was reported more than a decade ago. IFN-I exacerbates nephritis progression, and mice deficient in IFNAR1 develop a less severe phenotype upon cGN induction[53]. Our study corroborates these findings and further identifies IFN-I as a driver of T cell cytotoxicity. Cytotoxic T cells play crucial roles in cGN progression. The infiltration of CD8+ T cells into nephritic kidneys is a well-documented phenomenon in both human and experimental models[15,54–56], and their depletion was shown to improve outcomes in experimental GN models[54,57]. CD8+ T cells are capable of directly damaging endothelial and epithelial

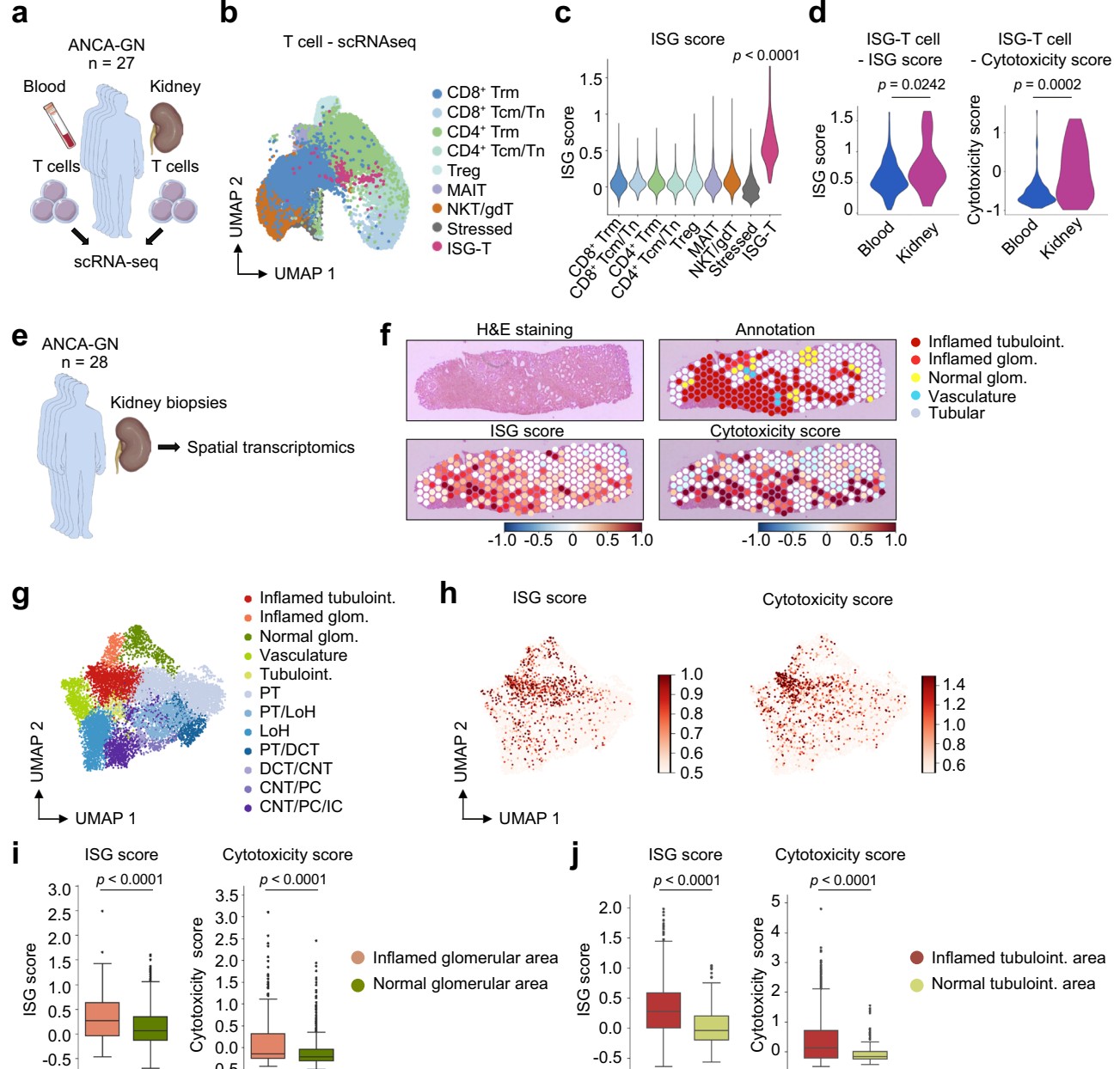

**Fig. 8 | scRNA-seq and sequencing-based spatial transcriptomics analyses reveal elevated ISG and cytotoxicity scores in kidney ISG-T cells and in inflamed kidney compartments. a** T cells were isolated from the blood and kidney biopsies of ANCA-GN patients for scRNA-seq analysis. **b** UMAP plot showing T cell clusters in ANCA-GN. **c** Violin plot depicting the ISG score levels in each T cell cluster. **d** Violin plots showing the levels of ISG and cytotoxicity scores in blood and kidney T cells. **e** Kidney biopsies from ANCA-GN patients were analyzed with the sequencing-based spatial transcriptomics technique. **f** Representative images of the hematoxylin and eosin (H&E)-stained kidney section (upper left), spatial distribution of annotated renal compartments overlaid on the H&E-stained kidney image (upper right), ISG score overlaid on the H&E-stained kidney image (lower left), and cytotoxicity score overlaid on the H&E-stained kidney image (lower right). UMAP plots illustrating annotated renal compartments across 10,763 spots from all spatial transcriptome slides (**g**) and the ISG and cytotoxicity scores (**h**). Box plots

showing ISG and cytotoxicity scores in glomerular (**i**) and tubulointerstitial (**j**) areas (Inflamed glom, n = 303; Normal glom, n = 507; Inflamed tubuloint, n = 1407; Tubuloint, n = 165). Box plots show the median (horizontal line inside the box), the 25th and 75th percentiles (lower and upper bounds of the box), and the whiskers extend to the most extreme data points not considered outliers (within 1.5 × IQR). Outliers beyond this range are shown as individual dots. **c** P value was calculated by the two-sided Wilcoxon rank-sum test. **d, i, j** P values were calculated by the unpaired two-tailed t-test with Welch's correction. Cell type abbreviations: ISG-T interferon-stimulated genes-expressing T, Trm tissue-resident memory T, Tcm central memory T, Tn naïve T, Treg regulatory T, MAIT mucosal-associated invariant T, NKT natural killer T, gdT γδ T, tubuloint tubulointerstitial, glom glomerular, LoH loop of Henle, PT proximal tubule, CNT connecting tubule, PC principal cell, IC intercalated cell, DCT distal convoluted tubule. Source data are provided as a Source Data file.

cells[13,58]. Furthermore, the rupture of Bowman's capsule permits CD8⁺ T cells to access the glomerular tuft and podocytes, facilitating the glomerular crescent formation[59]. In this study, our findings link the role of IFN-I and T cell cytotoxicity in cGN by identifying IRF7 as a marker of ISG-T cells and a potential molecular mechanism.

IRF7, a member of IRF family[30], has roles in IFN-I production in myeloid cells and fibroblasts[31]. *IRF7* knockout results in the loss of IFN-I production in mice and humans[60,61]. Although one study reported impaired expansion of virus-specific CD8⁺ T cells in *Irf7* knockout mice during lymphocytic choriomeningitis virus infection[32], the role of IRF7

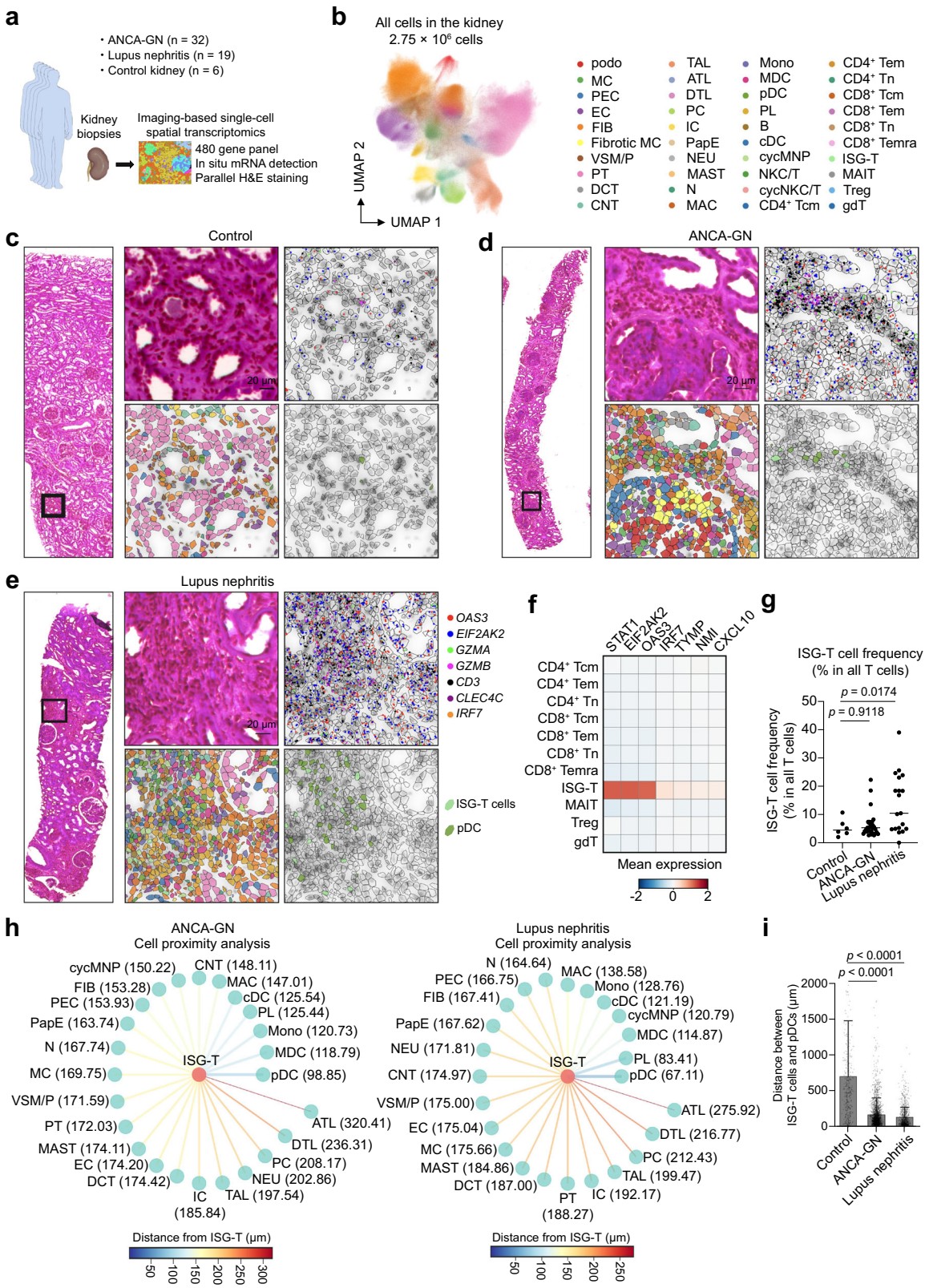

in T cell activation remains understudied. Although we were unable to obtain *Irf7* knockout mice, which presents a limitation of our study, our findings indicate that IRF7, at least in part, contributes to GzmB production in these cells.

In summary, our study identifies a novel T cell population, ISG-T cells, which are potentially induced by local IFN-I signaling in inflamed kidney tissues to efficiently expand the CTL population. These findings provide new insights into the role of IFN-I in the pathogenesis of autoimmune diseases.

## Methods

### Human studies

Human kidney tissues were obtained from individuals enrolled in the Hamburg GN Registry, the European Renal cDNA Bank, or the CRU 228

**Fig. 9 | Imaging-based single-cell spatial transcriptomics reveals the presence of ISG-T cells in close proximity to plasmacytoid dendritic cells in inflamed kidneys. a** Kidney tissue was analyzed using imaging-based single-cell spatial transcriptomics. **b** UMAP plot showing different cell clusters. **c–e** Representative images of kidney tissues under different conditions: H&E-stained tissue overview (far left), magnified view of the indicated area (top middle), DAPI-stained image overlaid with cell boundaries and specific marker genes (top right), segmented cells color-coded by cell type annotation using the same color scheme as in (**b**) (bottom middle), and segmented cells showing only ISG-T cells and pDCs in color with other cells shown in transparent (bottom right). These findings were confirmed in more than three areas and representative images were chosen. **f** Heatmap showing ISG expression levels in different T cell subtypes. **g** Dot plot showing the frequency of ISG-T cells among all T cells (Control, n = 6; ANCA-GN, n = 32; Lupus nephritis, n = 19). **h** Cell proximity analysis showing the median distances between ISG-T cells and different non-lymphocyte cell types. **i** Bar graph showing the distances

between ISG-T cells and pDCs in different conditions (Control, n = 414; ANCA-GN, n = 2429; Lupus nephritis, n = 952). P values were calculated by one-way ANOVA with Tukey's multiple comparison test. Cell type abbreviations: pDC plasmacytoid dendritic cell, podo podocyte, MC mesangial cell, PEC parietal epithelial cell, EC endothelial cell, FIB fibroblast, fib. MC fibrotic mesangial cell, PT proximal tubule, VSM/P vascular smooth muscle cell/pericytes, DCT distal convoluted tubule, CNT connecting tubule, TAL thick ascending limb of loop of Henle, ATL ascending thin limb of loop of Henle, DTL descending thin limb of loop of Henle, PC principal cell, and intercalated cell (IC) of collecting duct, PapE papillary tip epithelial cells abutting the calyx, NEU neuronal cell, N neutrophil, MAC macrophages, Mono monocytes, MDC monocyte-derived cell, PL plasma cell, cDC conventional dendritic cell, cycling mononuclear phagocyte, NKC/T natural killer cytotoxic T cell, cycNKC/T cycling natural killer cytotoxic T cell. Source data are provided as a Source Data file.

---

ANCA-GN cohort. In some cases, matched blood samples from the corresponding patients were also examined. The studies were approved by the *Ethik-Kommission der Ärztekammer Hamburg*, the local ethics committee of the Hamburg Chamber of Physicians (PV4806, Hamburg GN registry; PV5026, immune response in immune-mediated kidney diseases – ANCA; PV5822, Tumor nephrectomy), and conducted in compliance with the ethical principles outlined in the Declaration of Helsinki. Informed consent was obtained from each patient participating in the research. Patient data are available in the supplementary information.

### Animal experiments

All animal experiments were performed with the approval of the regulatory committee (Free and Hanseatic City of Hamburg, Authority for Justice and Consumer Protection, Office for Consumer Protection, Department of Food Safety and Veterinary Affairs) and carried out in accordance with German animal protection law. To ensure validity and reliability, mice were housed under specific pathogen-free (SPF) conditions. Experimental and control animals were bred separately after starting experiments. Male mice on a C57BL/6 background were used for experiments and were age-matched to minimize the variability. To induce experimental crescentic glomerulonephritis, mice aged 8–14 weeks were intraperitoneally injected with sheep serum targeting the glomerular basement membrane, following a methodology outlined in previous research[18,19,62]. For the IFNAR1 neutralization experiment, we injected 100 μg of InVivoMAb anti-mouse IFNAR-1 antibody (BioXCell, BE0241) or InVivoMAb anti-mouse IgG1 isotype control antibody (BioXCell, BE0083) on days 0, 3, 6, and 9 of the cGN. In the adoptive transfer experiments using *Ifnar1* knockout T cells, we isolated T cells from the spleen of B6(Cg)-*Ifnar1*[tm1.2Ees]/J mice (Strain #:028288). We transferred $2.5 \times 10^5$ CD4$^+$ T cells and an equivalent number of CD8$^+$ T cells into each *Rag1* KO mouse of the B6;129S7-*Rag1*[tm1Mom]/J strain (Strain #:002096). For analysis, the *Ifnar1* knockout group was compared to wild-type littermate controls. Mice were euthanized using isoflurane or carbon dioxide exposure followed by cervical dislocation, in accordance with institutional and national ethical guidelines.

### Histopathology, immunohistochemistry, and immunofluorescence

To assess glomerular crescent formation, 30 glomeruli for each mouse were analyzed blindly using Periodic acid-Schiff (PAS) staining on paraffin-embedded kidney sections[18,19,62]. For immunohistochemistry, paraffin-embedded kidney sections (2 μm) were stained with antibodies against CD3 (Cat# A0452; Dako), Gr1 (Cat# HM1039; NIMP-R14; Hycult Biotech, Uden), and Mac2 (Car# CL8942AF7; M3/38, Cedarlane, ON, Canada) to quantify leukocytes. For IF staining, primary antibodies against CD3 (Cat# A0452; Dako), CD4 (Cat# 100506; RM4-5, BioLegend), and CD8 (Cat# 100775; 53-67, BioLegend) were applied and

incubated at room temperature for 1 h. After washing, fluorochrome-conjugated secondary antibodies were used, and the fluorescence signals were captured using an LSM800 microscope with Airyscan and analyzed with Zen software (provided by Carl Zeiss, Jena, Germany). All antibodies were diluted 1/200.

### Isolation of human and murine leukocytes

Single-cell preparations were obtained from kidney and blood samples for human leukocyte analysis. The process involved enzymatic digestion of the kidney tissues with collagenase D at a concentration of 0.4 mg/ml (Cat# 11088858001; Roche, Mannheim, Germany) and DNase I at 10 μg/ml (Cat# AMPD1; Sigma-Aldrich, Saint Louis, MO) in RPMI 1640 medium at 37 °C for 30 min, followed by mechanical dissociation using the gentleMACS system (Miltenyi Biotec). Blood samples were processed for separation using Leucosep tubes (Cat# 227288; Greiner Bio-One, Kremsmünster, Austria). Subsequently, all samples were passed through a 30-μm mesh filter before antibody labeling and flow cytometric analysis.

Murine spleen cells were harvested by pressing the organ through a 70-μm cell strainer. Red blood cells were removed using a lysis buffer containing 155 mM NH$_4$Cl, 10 mM KHCO$_3$, and 10 μM EDTA at a pH of 7.2. For isolating lymphocytes from mouse kidneys, the organs were enzymatically digested with 400 μg/ml collagenase D (Cat# 11088858001; Roche, Mannheim, Germany) and 10 U/ml DNase I (Cat# AMPD1; Sigma-Aldrich, Saint Louis, MO) at 37 °C for 30 min, followed by mechanical dissociation using the gentleMACS system (Miltenyi Biotec). Then, leukocytes were isolated through density gradient centrifugation using 37% Percoll plus (Cat# GE17-5445-01; Merck Millipore) and a subsequent filtration step through a 30-μm cell strainer.

### Murine cell preparation for flow cytometric analysis

After blocking using mice serum, surface labeling of cells was carried out with fluorochrome-conjugated antibodies (CD45 (Cat# 103101; 30-F11; BioLegend), CD3 (Cat# 100311;145-2C11; BioLegend), CD4 (Cat# 100505; RM4-5; BioLegend), CD8 (Cat# 100711; 53-6.7; BioLegend), CD107a (Cat# 121611; 1D4B; BioLegend) along with a fixable dye for dead cells (Cat# L10119; Molecular Probes) to exclude non-viable cells from the analysis. For fixation and permeabilization, we used Cytofix/Cytoperm (Cat# 554714; BD Biosciences) following the manufacturer's guidelines. Antibodies targeted against GzmB (Cat# 515405; GB11; BioLegend) and IRF7 (Cat# 12-5829-82; MNGPKL; Invitrogen) were utilized for intracellular staining. Although fluorochrome combinations varied between experiments, the same antibody clone was consistently used for each target antigen. All antibodies were diluted 1/100.

### Human PBMC preparation for flow cytometric analysis

Peripheral blood samples were obtained from patients diagnosed with systemic lupus erythematosus (SLE). PBMCs were isolated using a

density gradient centrifugation method. Briefly, 20 mL of 70% Percoll solution (Cat# GE17-5445-01; Merck Millipore) was added to a 50 mL centrifuge tube, and 5 mL of whole blood was carefully layered on top. The tubes were centrifuged at $500 \times g$ for 30 min at room temperature. The PBMC layer was then collected from the interface between the Percoll solution and plasma. The isolated PBMCs were washed twice with complete culture medium by centrifugation at $350 \times g$ for 5 min. The purified cells were frozen in 10% DMSO + 90% FCS at −80 until analysis or immediately stained for flow cytometric analysis. Human PBMCs were stained with following fluorochrome-conjugated antibodies, (CD4 (Cat# 300501; RPA-T4; BioLegend), CD8 (Cat# 301002; RPA-T8; BioLegend), CD45RA (Cat# 983004; HI100; BioLegend), CD45RO (Cat# 304251; UCHL1; BioLegend), CD44 (Cat# 103001; IM7; BioLegend), CCR7 (Cat# 988902; G043H7; BioLegend), PD1 (Cat# 329933; EH12.2H7; BioLegend), CD62L (Cat# 980702; DREG-56; BioLegend), IRF7 (Cat# 656003; 12G9A36; BioLegend)). Although fluorochrome combinations varied between experiments, the same antibody clone was consistently used for each target antigen. All antibodies were diluted 1/100.

## Flow cytometry and cell sorting

Samples were measured with Symphony A3 (both BD Biosciences). Data was analyzed using the FlowJo software (Treestar, Ashland, OR). FACS sorting was conducted with an AriaFusion or AriaIIIu (BD Biosciences).

## In-vitro stimulation of cells

$2 \times 10^6$ cells were cultured in a volume of 2 ml of IMDM medium containing 10% FCS, streptomycin, and penicillin in a 24-well plate pre-coated with anti-CD3Ab (5 μg/ml; Cat# 100360; Biolegend), and 5 μg/ml anti-CD28 antibody was added (Cat# 102122; Biolegend). Mouse IFNb1 (50 ng/ml; Cat# 581302; BioLegend) or mouse IL-12 (50 ng/ml; Cat# 797204; BioLegend) were used during T cell stimulation.

## Retrovirus transduction

To overexpress non-functional and constitutively active variants of IRF7 or GFP control, the MIGR1 plasmid (created by Warren Pear, available through Addgene, plasmid #27490)[63] was utilized. Non-functional (1-456) and constitutively active (1-687 + 1264-1371) IRF7 variants were integrated into the MIGR1 vector previously prepared with EcoRI digestion. For retrovirus production, HEK293T cells were plated in 6-well plates and, on the following day, transfected with the constructed IRF7 or control vectors using the pCL-Eco plasmid (developed by Inder Verma, available through Addgene, plasmid #12371)[64] with Lipofectamine 3000 (Cat# L3000001; Invitrogen). Murine T cells, at a density of $2 \times 10^6$ cells per well, were activated in 24-well plates using anti-CD3 antibodies (2 μg/ml; Cat#100360; BioLegend) and anti-CD28 antibody (5 μg/ml; Cat #102122; BioLegend) for 2 days. Following this activation period, the culture medium was replaced with one containing the generated retrovirus and supplemented with polybrene. The cells were then centrifuged at $1000 \times g$ at room temperature for 1 h to enhance viral infection.

## RNA extraction and quantitative RT-PCR

RNA from the kidney was isolated with the NucleoSpin Kit (Cat# 740962.20; Macharey-Nagel, Düren, Germany) in compliance with the manufacturer's recommended protocol. Subsequently, the RNA (500 ng/sample) underwent reverse transcription using the High-Capacity cDNA Reverse Transcription Kit (Cat# 4374967, Thermo Fisher, Waltham, MA), and the StepOnePlus Real-Time PCR system (Thermo Fisher, Waltham, MA) was utilized for measurement. Primer sequence: Irf7 fwd (CCTCTGCTTTCTAGTGATGCCG), Irf7 rev (CGTAAACACGGTCTTGCTCCTG), Isg15 fwd (CATCCTGGTGAGG

AACGAAAGG), Isg15 rev (CTCAGCCAGAACTGGTCTTCGT). Stat1 (Mm01257286, Life Technologies), Stat5a (Mm03053818, Life Technologies), Stat5b (Mm00839889, Life Technologies) expression were measured with Taqman PCR system and 18s rRNA (Mm01168134, Life Technologies) was used as a house keeping gene.

## Single-cell RNA sequencing

Single-cell RNA sequencing was performed using single-cell suspensions prepared from human and mouse kidney tissues or blood. For the CITE-seq approach, monoclonal antibodies tagged with specific barcodes were used to label the cells (Biolegend, San Diego, CA). Cells positive for CD45 or CD3 markers were isolated through fluorescence-activated cell sorting (FACS) and then processed for droplet-based single-cell RNA analysis and transcriptome library construction employing the Chromium Single Cell 5´ Library & Gel Bead Kit v2 and following the protocols provided by 10x Genomics (Pleasanton, CA). The scRNA-seq libraries were sequenced using the Illumina NovaSeq6000 platform, utilizing a 100-cycle sequencing run.

## Alignment, quality control, and preprocessing of scRNA-seq data

Alignment, quality control, and initial processing of the scRNA-seq datasets followed established procedures. Specifically, the Cell Ranger software suite (version 5.0.1, from 10x Genomics) was employed for cellular barcode demultiplexing and aligning sequence reads to the reference genomes, specifically refdata-cellranger-hg19-1.2.0 for human samples and refdata-gex-mm10-2020-A for mouse samples. Seurat (version 4.0.2) software's HTODemux function was applied to sort out cells based on their hashing tags. Cells displaying fewer than 500 genes, more than 6000 genes, or a mitochondrial gene content exceeding 5% were excluded from further analysis. In the case of CITE-seq, a synthetic reference genome was created using the cellranger mkref utility provided by Cell Ranger. Alignment of CITE-seq data to this custom reference was achieved with the Cell Ranger count function. Antibody-derived tag (ADT) information was then merged with the single-cell RNA sequencing data via Seurat. The detailed methodology was explained in previous publication[18].

## Dimensionality reduction, clustering, enrichment analysis, and scores

The Seurat package (version 4.0.2) was used for unsupervised clustering to analyze single-cell RNA sequencing data[65]. The process began with normalizing gene counts for each cell against the total library size, followed by log transformation. Seurat's version 4 integration methods (using functions FindIntegrationAnchors and IntegrateData, with dimensions set to 1:30) were applied to mitigate batch effects. The resulting integrated data were scaled using the ScaleData function, adhering to default settings. Dimensionality reduction was achieved through principal component analysis (PCA) on the scaled dataset, employing the RunPCA function with a specified number of principal components (npcs = 30). Clusters were visualized using Uniform Manifold Approximation and Projection (UMAP). To identify the most significantly differentially expressed genes across clusters, the FindAllMarkers function was used (setting the minimum percentage of cells expressing a gene at 0.1). The Wilcoxon rank sum tests were used for this purpose. The FindMarkers function (also set at min.pct = 0.1) facilitated differential expression analyses between clusters or specific groups by the Wilcoxon rank sum tests. We annotated clusters based on the differentially expressed genes and typical marker genes[18,66–71]. For broader biological insight, enrichment analyses were performed using the metascape platform[72]. To analyze scores (ISG score and cytotoxicity score) in scRNA-seq data, we used Seurat function AddModuleScore, which calculates scores by averaging the expression levels of a predefined gene set and subtracting the average expression

of a randomly selected set of control genes. To assess cell differentiation, we performed trajectory analysis using Monocle 3[27], designating the naïve T cell cluster as the root (starting point of differentiation) to calculate pseudotime scores.

## Sequencing-based spatial transcriptomics analysis

For spatial transcriptomics (ST), formalin-fixed paraffin-embedded (FFPE) tissue sections of kidney biopsies from ANCA-GN patients were transferred on Visium (10x Genomics) slides (spatial for FFPE gene expression human transcriptome) and processed following the manufacturer's instructions. Next-generation sequencing was performed using an Illumina NovaSeq 6000 aiming at 25,000 reads per spot (PE150). For alignment, the human genome assembly GRCh38-2020-A was used. Mapping to the genome was performed using 10x Genomics *Space Ranger* (v2.0.1). The ST gene expression data was analyzed using Scanpy[73] (v1.9.3) in Python (v3.9.7). The following parameters in Scanpy's preprocessing pipeline were used to filter poor-quality spots: min_genes = 100, min_spots = 3, min_counts = 2000, max_counts = 35000. The filtered spot counts were normalized to sum to 10,000, and data was log2-transformed with a pseudo-count of 1. For clustering and annotation, principal components (n_comps=50) were computed on the highly variable genes (highly_variable_genes in Scanpy with default settings and slide-name as batch_key). The batch effect corresponding to the slide was removed using Harmony[74] (v0.1.0) in R (v4.1.1). To identify clusters, Leiden clustering (scanpy.tl.leiden) was performed on Uniform Manifold Approximation and Projection (UMAP) data projections with a resolution of 1.2. The UMAP projections were generated on a neighborhood graph constructed using scanpy.pp.neighbors with n_neighbors = 10. Cluster annotations were performed using the following cell type-specific markers from a reference kidney single-cell dataset[75]—proximal tubules (PT): LRP2, CUBN, SLC13A1, distal convoluted tubules (DCT): SLC12A3, CNNM2, FGF13, KLHL3, LHX1, TRPM6, connecting tubules (CNT): SLC8A1, SCN2A, HSD11B2, CALB1, principal cells (PC): GATA3, AQP2, AQP3, intercalated cells (IC): ATP6V0D2, ATP6V1C2, TMEM213, CLNK, ascending thin loop of Henle (thin limb): CRYAB, TACSTD2, SLC44A5, KLRG2, COL26A1, BOC, thick ascending loop of Henle (TAL): CASR, SLC12A1, UMOD. Endothelial cells (Endo): CD34, PECAM1, PTPRB, MEIS2, EMCN, vascular smooth muscle cells (vSMC)/pericyte: NOTCH3, PDGFRB, ITGA8, fibroblasts: COL1A1, COL1A2, C7, NEGR1, FBLN5, DCN, CDH11, podocytes: PTPRQ, WT1, NTNG1, NPHS1, NPHS2, CLIC5, PODXL, immune cells: PTPRC, CD3D, CD14, CD19.

## Imaging-based spatial transcriptomics analysis

For imaging-based spatial transcriptomics analysis, 5 μm thick human kidney FFPE sections were placed on Xenium slides and processed according to manufacturer's instructions (10x Genomics, Pleasanton, CA, USA). Subsequently, HE staining was conducted, and whole slide images were obtained on a Leica DMi8 system using the Leica Application Suite X software Version 3.7.4.23463 software (Leica, Wetzlar, Germany). For the optimal cell segmentation, the default segmentation was refined using Baysor (version 0.6.2) with confidence in prior probability set to 0.5. The resulting datasets were log-normalized using *scanpy.pp.normalize_total* and *scanpy.pp.log1p* methods from Scanpy (version 1.10.1). The target_sum was set to 1000. LogisticRegressionCV class from scikit-learn (version 1.3.1) was used to train for the classification task with input being the normalized count matrix of the reference data and target being the cell type labels. To analyze the spatial relationships between ISG-T cells and other cell types in Xenium data, we calculated the minimum Euclidean distances (in μm) between each non-ISG-T cell and its nearest ISG-T cell using spatial.distance_matrix function from Python package Scipy (v1.11.3). The analysis was performed separately for each biopsy sample. The median distances were then visualized as a network graph using Python package networkx (v3.2.1), where ISG-T cells were positioned at the center

and other cell types were arranged radially with their distances represented by edge colors (blue to red) and widths (thicker edges indicating closer proximity). For more details, please refer to our previous manuscript[39].

## Statistics

Statistical analysis was conducted using GraphPad Prism (La Jolla, CA). The results are shown as mean + SEM when presented as a bar graph or as single data points with the mean in a scatter dot plot. Differences between the two individual groups were compared using the Mann-Whitney test or two-tailed t-test. In the case of three or more groups, a one-way ANOVA with Tukey's multiple comparisons test or the Kruskal-Wallis test followed by Dunn's multiple comparisons test was used. The correlation coefficient r was calculated using a Pearson correlation, and the corresponding P value was based on a t-distribution test. All experiments were repeated three times or more except for omics studies in which multiple clinical samples were included.

## Reporting summary

Further information on research design is available in the Nature Portfolio Reporting Summary linked to this article.

## Data availability

Public datasets: scRNA-seq data of blood T cells from patients with AAV is available at Genomic Expression Archive (GEA) with accession code E-GEAD-635 [https://humandbs.biosciencedbc.jp/en/hum0416-v1][17]. scRNA-seq data of blood T cells of SLE patients is available in the dbGaP database under phs002048.v1.p1 [https://www.ncbi.nlm.nih.gov/projects/gap/cgi-bin/study.cgi?study_id=phs002048.v1.p1][20]. The human transcriptome data of CD8+ T cells stimulated with IFN-α is available in NCBI Gene Expression Omnibus (GEO) under GSE17302[24]. The transcriptome data of spleen CD4+ T cells and CD8+ T cells from mice treated with IFN-α is available in NCBI GEO under GSE75202[25]. Transcriptome data of kidney biopsies from ERCB are available in the NCBI GEO under GSE104954[36]. scRNA-seq datasets of blood T cells from patients with SARS-CoV2 and healthy donors are available in NCBI GEO under GSE163668[49].

Publicly available datasets previously reported from our group: Our scRNA-seq datasets of CD45+ leukocytes (CD3+ and CD3- cells) isolated from murine kidneys are available in the NCBI GEO under GSE200880[19]. Our scRNA-seq dataset of blood and kidney T cells from patients with ANCA-GN is available in NCBI GEO under GSE253633[71]. Our sequencing- or imaging-based spatial transcriptomics datasets[39,71] are available in NCBI GEO under GSE250138 and GSE294965.

Newly generated datasets for this study: The scRNA-seq datasets of CD3+ cells from healthy murine kidneys, CD45+ cells from the blood and spleen at day 10 of cGN, kidney cells from wild-type B6 mice at day 10 of cGN generated in this study are available in NCBI GEO under GSE296304.

All data are included in the Supplementary Information or available from the authors. The raw numbers for charts and graphs are available in the Source Data file whenever possible. Source data for graphs are provided with this paper. Source data are provided with this paper.

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

## Acknowledgements
FACS sorting was performed at the UKE FACS sorting core facility. Single-cell RNA sequencing was performed at the UKE Single-Cell Core Facility. Graphical images were produced using Servier Medical Art images (https://smart.servier.com). We gratefully acknowledge Dr. Ludmilla Unrau from the Bernhard Nocht Institute for Tropical Medicine for harvesting and providing spleens from *Ifnar1* KO mice and their littermates for our experiments. This study was supported by grants from the *Deutsche Forschungsgemeinschaft* (DFG) to U.P. (SFB 1192 A1 and C3). N.A. was supported by a Research Fellowship of the Japan Society for the Promotion of Science.

## Author contributions
Conceptualization: N.A. Methodology: H.W., N.A., H.J.P., J.E., Z.S., S.L.J.S., A.P., A.K., and U.P. Formal analysis: H.W., N.A. In-vitro analysis: H.W. and N.A. Flow cytometry: H.W. and N.A. scRNA-seq data analysis: N.A. Spatial transcriptomics data analysis D.P.S., R.K., and S.B. Data analysis: H.W., N.A., A.K., and A.P. Renal histology: N.A. and U.P. Patient cohorts: J.E., T.B.H., C.F.K., and U.P. Writing the manuscript: H.W., H.W.M., N.A., and U.P. Visualization: H.W. and N.A. Funding acquisition: C.F.K. and U.P.

## Funding

## Competing interests
The authors declare no competing interests.
