## [Transparent Peer Review file · Nature Communications]

Type I interferon drives T cell cytotoxicity by upregulation of interferon regulatory factor 7 in autoimmune kidney diseases

Corresponding Author: Dr Nariaki Asada

Version 0:

Reviewer comments:

Reviewer #1

(Remarks to the Author)

Wang and coworkers present a newly described T cell population in their manuscript, which can be found in increasing frequencies in patients with either ANCA vasculitis or Lupusnephritis. Those T cells can be either CD4+ or CD8+, have a cytotoxic character and develop via stimulation of type 1 IFN and IRF7, which leads to Granzyme B production. The manuscript works largely with single cell RNAseq data, but also includes in vitro and an in vivo mouse model to prove the function of this cell population. I find merit in the study since defining a new immune cell population is always difficult, but I feel that additional wet lab work is needed to really prove their existence and to describe them and their function better.

My major comments are:

1. The characterization of this cell population only relies on single cell RNAseq data. Since they can be detected in the peripheral blood, it would be important to characterize them better (memory marker, exhaustion markers, etc). compared to healthy individuals? Can they be detected by flow cytometry using a specific antibody (IRF7) also in humans?
2. Also, localization in kidney tissue would help to convince me not only about their existence but also their localization in the kidney. It would be important to give some hints about their localization in the kidney of patients (or mice) with GN and how they interact with other infiltrating immune cells (eg Tregs). I do not believe that special transcriptomics is really the best method to show localization of immune cells.
3. It has already been shown in 2020 that GzmB producing CD4 T cells exist in SSc, IgG4-RD, and primary Sjogren syndrome patients. Are these the same cells as described by the authors? Maybe single cell data from SSc, IgG4-RD or primary Sjogren syndrome patients exist and can be compared to the SLE and ANCA patients (Maehara T., N.Kaneko, C. A.Perugino, H.Mattoo, J.Kers, H.Allard-Chamard, V. S.Mahajan, H.Liu, S. J.Murphy, M.Ghebremichael, et al 2020. Cytotoxic CD4+ T lymphocytes may induce endothelial cell apoptosis in systemic sclerosis. J. Clin. Invest. 130: 2451–2464.)
4. The authors are very vague about the function and role of these new cell population(s).
5. I do not entirely understand why the authors claim that these cells develop in the kidney, while they find them in the peripheral blood of the patients? No data from peripheral blood of nephritic mice is shown.
6. What time point after induction of nephritis has been evaluated in Figure 6? It would help to also show peripheral blood data.
7. It would help to show CD4+GzmB+ and CD8+GzmB+ T cell dynamics during the mouse model in various organs (sec lymphoid organs and kidney).
8. Anti-IFNR1 treatment has been started together with the induction of nephritis. Why was this setting used and how was the dose evaluated? This treatment is rather unspecific to really test the hypothesis. Data showing immune cell infiltration into the kidney is sparse. How are T cells, neutrophils and macrophages infiltrating the kidney? Why is there no difference in CD4+GzmB+ T cells and in albuminuria (Supplemental Figure)?

Reviewer #2

(Remarks to the Author)

The study highlights the role for IFN-I in driving T cell cytotoxicity and tissue damage in ANCA-associated vasculitis (AAV) and systemic lupus erythematosus (SLE), the mechanisms underlying T cell activation remain inadequately explained. There are concerns about deficiencies in the present work that limit enthusiasm for publications. Specifically:

1. Throughout the article, there appear to be several instances of over-interpretation of the research findings, leading to conclusions that may seem somewhat speculative. For example, in line 95, there is a lack of data analysis or experimental evidence supporting the activation and differentiation of T cells stimulated by type I interferon into ISG-T/ISG-CTL. Moreover, there is no analytical data provided to confirm whether ISG-CTL represents an intermediate phenotype between ISG-T and CTL.
2. The expression patterns of marker genes used to define various cell types in this study appear unclear. For instance, Fig 1C and 2E indicate no notable expression of highly specific genes in CD4+TEM. Furthermore, the CTL cells in the ANCA dataset do not exhibit significant expression levels of CD4 and CD8A, whereas CTL cells in the Lupus dataset show high expression of CD8A. Could you clarify if these two cell types are considered identical? Additionally, what is the rationale for separately analyzing the ANCA dataset from the Lupus dataset?
3. The number of patient samples exceeds that of healthy donors (HD). Are the patient cell counts also significantly higher than those of HD? If so, how can we determine that ISG-T and other cells have expanded based on that ANCA or SLE cells make up more than 50% of the total? Could statistical analysis be included to substantiate the expansion of ISG-T and other cells? Additionally, could the ISG score in Figure 1D be presented with statistical analysis?
4. The Figure legend throughout the article do not specify the statistical analysis methods corresponding to the figures.
5. As depicted in Figure 2E, GZMB and GZMA exhibit higher expression levels in CD8 TEM cells. Meanwhile, Figure 2F indicates that GZMB and GZMK are not predominantly expressed in ISG-T cells, Figure 2G shows that only a small subset of cells with elevated ISG scores also demonstrate high cytotoxicity scores. Thus, further investigation is required to determine whether ISG-T cells exhibit a cytotoxic phenotype. The co-expression of ISG genes and cytotoxicity-related genes can be visualized in these data, and multi-color tissue immunofluorescence can be utilized to verify the co-expression of ISG and cytotoxicity-related genes in kidney tissues of the cGN mouse model.
6. As is widely recognized, Monocle3 can designate the starting point of differentiation trajectories. The authors specified naive T cells as the starting point. Was the starting position of ISG-T manually specified? The research method does not clarify the basis for calculating the pseudotime score. Besides Monocle3, there are numerous other methods such as RNA velocity analysis capable of conducting pseudotime trajectory analysis. Can these alternative methods lead to similar conclusions?
7. In addition to GZMB, there are many other indicators for detecting cytotoxicity levels, such as the expression of CD107a, etc. The expression level of GZMB alone cannot lead to the conclusion that IFN-I drives T cell cytotoxicity (line 156).
8. As depicted in Figure 4B, STAT1 exhibits high average expression in ISG-T, and also plays a crucial role in T cell differentiation. Has there been an investigation into whether the expression level of STAT1 changes after IFN-I neutralization, and whether STAT1 expression correlates with GZMB expression? Additionally, how does IRF7 regulate the levels of GZMB?
9. In lupus nephritis or cGN mouse model, ISGs are highly expressed not only in CD4 and CD8 T cells, and the inhibition of IFNAR1 antibody at disease onset may not solely affect ISG-T. Modeling CD4- and CD8-specific knockout of IRF7 or IFNAR receptors would provide clearer insights into how IFN-I influences disease progression through regulation of ISG-T.
10. Is IRF7 elevated in infection models, and then is there a regulatory report that increases GZMB expression?
11. In the spatial transcriptome data of ANCA-GN patients, do the regions with high interferon scores and cytotoxicity scores correspond to CD4 or CD8 T cells? Do these regions exhibit high expression of the IRF7 gene?
12. There is potential for improvement in thesis writing, particularly with managing English tenses.

Reviewer #3

(Remarks to the Author)

This manuscript explores the ability of type I interferons (IFNs) to drive T cell cytotoxicity and subsequent renal damage within ANCA associated vasculitis (AAV) and systemic lupus erythematosus (SLE), systemic autoimmune diseases that involve glomerulonephritis. Using RNAseq data, this study has identified a unique subset of T cells characterised by the expression of type I IFN genes, ISG-T cells, that are expanded in AAV and SLE. They show that these subsets are present in mouse models of crescentic glomerulonephritis (cGN) and are associated with cytotoxicity through IRF7. They also show a correlation between interferon-stimulated gene (ISG) expression and genes related to cytotoxicity in the ISG-T subsets from SLE and AAV human kidney samples. This work offers novel insights into the role of type I IFN mediated T cell toxicity in autoimmune diseases involving renal inflammation and I commend the authors for their thorough and interesting work.

However, there are some extra details that are required to support some of the interpretations made by the authors and thus fully support the conclusions of this work:

1. A detailed participant table for all human samples is required for appropriate interpretation of the data provided. Factors such as age, sex, medication, disease activity/severity at the time of sampling and co-morbidities, can all have dramatic effects on gene expression. Particular factors, such as age and sex, may need to be included in your statistical models to ensure that the changes that you describe are disease specific. This is required for data presented in Figures 1, 7 and 8.
2. A major issue is that the authors have not really proven mechanism with regard: (i) that in patients or their mouse model, that IRF7 is the definitive transcription factor needed for ISG-T cell development and (ii) a mechanism for how IRF7 is regulating granzyme B (or A or K) expression in patients and/or their model. Point (i) would probably require knock out animals and thus, likely beyond the scope of this work, but I think it is very important that the authors word their interpretation of what they have done and highlight the limitations/alternatives. For point (ii), this could be explored further. A possibility could be the IRF7 is promoting/amplifying further IFN-I responses, which in T cells, could be working via IFN-I driven STAT5 activation to drive GZMB expression.
3. The "Lupus Nephritis" patients analysed in Figure 7, did they have cGN as their kidney pathology?

4. Can you clarify heat maps in Figure 1 C whether these are comparing disease samples to HD or whether these compare between T cells subtypes? Can you also make the figure scales the same for the ANCA data set and SLE data set. How were the marker genes shown in Fig 1C chosen?
5. Page 5, line 90-91 “The ISG-T cell population expanded in the blood of patients versus healthy donors, suggesting an IFN-I mediated activation of T cells in autoimmune disease” – this is clear from the graphs in SLE but less so in AAV. Similarly, on page 6 line 123-124 the differences are unclear. Can you include percentages in the text? I would also suggest including a section in the discussion regarding the consistently lower IFN expression within AAV samples compared to SLE (Figure 1 and Figure 7) with respect to current literature in the field.
6. What were the n numbers for mice used in the in vivo studies?
7. It is very reassuring to see the validation of the results from Figure 2 from the cGN (nephrotoxic nephritis model) mouse model in an additional model more relevant for lupus nephritis (the MRL/lpr lupus prone nephritic model results in Supp Fig 3) and really helps support the conclusions of the authors; can you do similar on a relevant ANCA-related kidney disease mouse model dataset?
8. In Figure 5, how similar to health control mice are the measures following administration of anti-IFNAR1 ab?
9. In patient included in Figure 1, did any of these have renal disease involvement and in particular, cGN?
10. I would strongly suggest rethinking the wording/conclusion of lines 221 –223 and associated conclusions. I think this is associated with the fact that there is not enough attention paid throughout about the fact that ultimately, AAV and SLE are quite different diseases and in relation to IFN-I dysregulation, although very clear in SLE pathology, both systemically and in the kidney, in AAV, this IFN-I phenotype is strong in kidney (pt 24 below) and murine models (<https://rupress.org/jem/article/219/10/e20220759/213416/Monocyte-derived-macrophages-aggravate-pulmonary>), but less conclusive systemically, particularly studies that use properly age/sex matched healthy and disease cohorts, with appropriate positive and negative disease controls (<https://nature.com/articles/s41598-021-87760-4>).
11. I would disagree that Supp 5 B & C support the conclusion in results text lines 184 –186 regarding improvement of proteinuria and BUN levels.
12. I don't think the analysis presented in Figure 6 and Supp Fig 6 is supportive enough of the overall conclusion of that results section, that “IFN-I signaling in the inflamed kidney tissue is mediated by the cross-talk between myeloid cells and T cells”. I think the data presented is supportive of a hypothesis that is might be mediated via this mechanism, but the authors would need to present more experimental work/analysis to support that as a conclusive mechanism, so either that needs to be done or else that section toned down, particularly the section heading.
13. Due to small differences in overall percentages noted with respect to GzmB, I would suggest adding the percent into the results text as the scale for these graphs can be misleading when not set to 100%.
14. I suggest you add Supplemental Fig. 5A to Fig. 5. (CD4 GzmB data). Although not significant, it still supports your overall message and keeps consistency across figures.
15. There seems to be no gradients present in Fig. 2 G, Fig. 3 B and E, Fig. 6 G, Supplementary Fig 3. E and G, Supplementary Fig. 6. Was this on purpose? Can you make these more clear, maybe by adding multi-coloured gradients i.e. from white to orange to pink? I would also like to see consistency across the scales for the UMAPs. Some of these go from 1-3 some go from 0-2 etc. This is particularly important in Fig. 3.
16. Please increase the font size of supplemental figure 1 G-H scale.
17. With regards to point 1, it appears that the SLE patients analysed in Fig. 7 B may cluster into 2 separate subgroups based on the IFN-I associated genes and the authors may wish to comment on the heterogeneity of this cohort/human cohorts.
18. Please make the scales for the SLE and ANCA data in Figure 7 C the same. The HD look different on each graph raising concerns on reliability of data though I think this is due to the scaling.
19. Can you include the quantity of RNA used for PCR work in the methods section?
20. Smaller data sets within this study should be statistically analysed using non-parametric tests. Where this has not been done, I would suggest including a comment on normality within the methods.
21. Can you include more detail on the ISG and cytotoxicity score calculation in the methods? A score of -0.5 for HD was not intuitive for me as a reader.
22. Note typo “Healthy” in Fig 1A.
23. Certain papers have shown increased ISG expression within the kidneys of SLE and AAV patients. Neutrophils and monocytes have also been shown to be major sources of these transcripts within SLE patients, which does not agree with Fig. 6 within this manuscript that only noted DC/macrophages as the source of ISG expression within kidney samples. Please include this in your discussion.
<https://www.ncbi.nlm.nih.gov/pmc/articles/PMC8318865/> - Chen et al. eBioMedicine 2021
<https://www.ncbi.nlm.nih.gov/pmc/articles/PMC2760083/pdf/nihms142880.pdf> - Kessenbrock et al. Nat Med 2009
24. Please add a section within the discussion as to possible reasons why there is no upregulation of granzyme expression in blood RNAseq data (Fig.1) versus cytotoxicity in kidney (Fig 7/8).
25. It is noted that the authors only use male mice for their in vivo models of cGN (the nephrotoxic nephritis model), which in the context of the cGN model produces a more severe form of disease. Ideally moving forward, and for more translational relevance given the significant sex differences in the disease the authors focus on, I would strongly suggest the authors include female mice in future work.

Reviewer #4

(Remarks to the Author)

Version 1:

Reviewer comments:

Reviewer #1

(Remarks to the Author)

Congratulations to the authors for this successful, intensive revision of the manuscript and I am happy with the additional experiments and changes. I have no further questions or comments.

Reviewer #2

(Remarks to the Author)

I appreciate the efforts made by the authors in revising the manuscript based on my previous comments. The revised version has improved in several aspects. However, there are still some limitations that need to be addressed to further substantiate the authors' conclusions. Below are my further comments and suggestions for additional refinements.

1. In the part of 'Introduction', the author describes CTLs as mature cells that develop following T cell activation. However, the analysis results suggest that ISG-T cells possess cytotoxic functions despite being immature. This appears to be a contradiction. Could the author clarify this inconsistency?
2. The authors propose that IFN- γ induces IRF7 expression, however, the mechanism by which IRF7 regulates GZMB expression remains insufficiently supported by evidence. Given that STAT1 is also highly expressed in ISG-T cells, what was the rationale for prioritizing IRF7 for validation? Furthermore, IRF7 overexpression leads to an upregulation of STAT1—does IRF7 modulate ISG cytotoxicity through STAT1, or does it directly regulate GZMB expression?
3. Can the spatial transcriptomics data reveal the co-localization of ISGs and GZMB expression?
4. Is the percentage in line 87 representing the mean or another metric? This should be clearly annotated.
5. The designation of ISG-T as an immature state in Results 2 is not sufficiently substantiated. Although scCITE-seq data is provided, the absence of direct flow cytometry validation weakens the evidence supporting this classification. Moreover, the conclusions in the second and fourth sections of the results are redundant. Would it be advisable to consider merging them for improved clarity and conciseness?
6. In Supplementary Figure 4h, there is no evident positive correlation between the cytotoxicity score and the ISG score. Additionally, the R^2 values are not indicated.
7. Figure 2 already presents the gene expression and cytotoxicity scores of ISG-T in the cGN model, yet Figure 3 also focuses on renal T cells in the cGN model. What is the rationale for separately reanalyzing CD4⁺ and CD8⁺ T cells?
8. Additionally, in Supplementary figure 11b, the ISG-CTL population in SLE exhibits a high pseudotime score, suggesting it represents a terminally differentiated, mature T cell subset. This appears inconsistent with the pseudotime trajectory analysis conducted separately in the cGN model. How can this discrepancy be explained?
9. The pseudotime trajectory analysis does not explicitly indicate trajectory direction. How, then, can it be supposed that ISG-T represents an intermediate population in the differentiation process from naive T cells to mature CTLs?(line 173-174)
10. Does IRF7 knockout affect IFN- β -induced cytotoxicity and ISG expression in CD4⁺/CD8⁺ T cells? Additionally, was ISG expression assessed in IRF7 overexpression system?
11. In Figure 6, is ISG expression selectively elevated in T cells within the cGN model, or is it also upregulated in other cell types? Similarly, is IRF7 expression restricted to CD4⁺ and CD8⁺ T cells, or is it observed in additional immune populations? Furthermore, the initial patient analysis identified ISG-T in the peripheral blood of ANCA and SLE patients. Why, then, are GZMB⁺ CD4⁺/CD8⁺ T cells absent in both the peripheral blood and spleen of cGN mice?
12. Spatial transcriptomics data indicate that the cell type closest to ISG-T is pDC. What is the nature of the interaction between pDCs and ISG-T? Are there specific ligand-receptor interactions or cytokine signaling pathways mediating their crosstalk?

Reviewer #3

(Remarks to the Author)

The authors have adequately addressed all my comments. Their changes to this paper in response to reviewers' comments significantly strengthens the insights into novel T cell functionality in autoimmune kidney diseases.

Version 2:

Reviewer comments:

Reviewer #2

(Remarks to the Author)

The authors have response to my questions. I have no further comment.

Reviewer #1:

Remarks to the Author:

Wang and coworkers present a newly described T cell population in their manuscript, which can be found in increasing frequencies in patients with either ANCA vasculitis or Lupusnephritis. Those T cells can be either CD4+ or CD8+, have a cytotoxic character and develop via stimulation of type 1 IFN and IRF7, which leads to Granzyme B production. The manuscript works largely with single cell RNAseq data, but also includes in vitro and an in vivo mouse model to prove the function of this cell population. I find merit in the study since defining a new immune cell population is always difficult, but I feel that additional wet lab work is needed to really prove their existence and to describe them and their function better.

We thank the reviewer for the constructive and insightful comments.

Comments:

1. The characterization of this cell population only relies on single cell RNAseq data. Since they can be detected in the peripheral blood, it would be important to characterize them better (memory marker, exhaustion markers, etc). compared to healthy individuals? Can they be detected by flow cytometry using a specific antibody (IRF7) also in humans?

In response to the reviewer's suggestion, we aimed to further characterize ISG-T cells based on surface markers. As a first step, we investigated surface protein levels by analyzing our single-cell Cellular Indexing of Transcriptomes and Epitopes by Sequencing (scCITE-seq) data of ANCA-GN patients. Using the barcoded antibodies, this approach enabled us to evaluate mRNA expression and surface protein levels at single-cell resolution. Following unbiased clustering and annotating ISG-T cells based on their transcriptomic profile, we analyzed the surface protein levels. As a result, we found that ISG-T cells were CD45RA⁻, CD44⁺, CD62L⁺, CD69⁻, and CD279/PD1⁻. We did not observe a clear difference in the ISG-T-cell phenotype between ANCA-GN and control conditions (data not shown). The results are included in the revised Supplementary Fig. 3a-c.

scCITE-seq analysis of T cells. **a** T cells from ANCA-GN patients and healthy donors were analyzed with scCITE-seq. **b** UMAP plot showing different clusters. **c** Heatmap illustrating each cluster's surface protein levels and marker gene expression.

For SLE, we analyzed blood T cells from three patients using flow cytometry. While IRF7 production was not observed in T cells from healthy donors, we identified IRF7-positive ISG-T cells in patients with SLE. These ISG-T cells were characterized as CD45RA^{±/±}, CD45RO^{-/±}, CD44^{-/±}, CCR7^{-/±}, CD69⁻, and CD279/PD1^{-/±}. The results are included in the revised Supplementary Fig. 3d-g.

Flow cytometry analysis of blood ISG-T cells from SLE patients. **a** T cells from SLE patients were analyzed with flow cytometry. Representative flow cytometry plots showing IRF7-positive ISG-T cells. **b-d** Heatmaps illustrating the expression of surface markers across different blood T-cell subsets in SLE patients. Collectively, these findings support that ISG-T cells are in the early stages of differentiation, as suggested by our pseudotime analyses.

2. Also, localization in kidney tissue would help to convince me not only about their existence but also their localization in the kidney. It would be important to give some hints about their localization in the kidney of patients (or mice) with GN and how they interact with other infiltrating immune cells (eg Tregs). I do not believe that special transcriptomics is really the best method to show localization of immune cells.

Following the reviewer’s comment, we sought to investigate the localization of ISG-T cells in the kidney tissues of patients with ANCA-GN or lupus nephritis. Due to the lack of perfect markers of ISG-T cells for immunofluorescence staining and the low resolution of our Visium spatial transcriptomics data, we established an imaging-based single-cell transcriptomics technique based on the Xenium platform. This analysis successfully identified the ISG-T cell population in the kidney tissues. Cell proximity analysis revealed the presence of ISG-T cells in close proximity to plasmacytoid dendritic cells, the major producers of IFN-I, among non-lymphocyte cell types. Among T cells, ISG-T cells were closely localized to naïve T cells, potentially because ISG-T cells differentiate from naïve T cells. These data are presented in the revised Fig. 9 and Supplementary Fig. 10.

Imaging-based single-cell spatial transcriptomics reveals the presence of ISG-T cells in close proximity to plasmacytoid dendritic cells in inflamed kidneys. **a** Kidney tissue was analyzed using imaging-based single-cell spatial transcriptomics. **b** UMAP plot showing different cell clusters. **c-e** Representative images of kidney tissues under different conditions: H&E-stained tissue overview (left), magnified view of the indicated area (upper middle), DAPI-stained image overlaid with cell boundaries and specific marker genes (upper right), segmented cells color-coded by cell type annotation using the same color scheme as in **b** (lower middle), and segmented cells showing only ISG-T cells and pDCs in color with other cells shown in transparent (lower right). **f** Heatmap showing ISG expression levels in different T cell subtypes. **g** Dot plot showing the frequency of ISG-T cells among all T cells. **h** Cell proximity analysis showing the median distances between ISG-T cells and different non-lymphocyte cell types. **i** Bar graph showing the distances between ISG-T cells and pDCs in different conditions. Error bars represent the standard deviation. P values were calculated by one-way ANOVA with Tukey's multiple comparison test (* $p < 0.05$, **** $p < 0.0001$). For cell type abbreviations, please see figure legend for Fig. 9

3. It has already been shown in 2020 that GzmB producing CD4 T cells exist in SSc, IgG4-RD, and primary Sjogren syndrome patients. Are these the same cells as described by the authors? Maybe single cell data from SSc, IgG4-RD or primary Sjogren syndrome patients exist and can be compared to the SLE and ANCA patients (Maehara T., N.Kaneko, C. A.Perugino, H.Mattoo, J.Kers, H.Allard-Chamard, V. S.Mahajan, H.Liu, S. J.Murphy, M.Ghebremichael, et al 2020. Cytotoxic CD4+ T lymphocytes may induce endothelial cell apoptosis in systemic sclerosis. *J. Clin. Invest.* 130: 2451–2464.)

As the reviewer pointed out, it was shown that CD4⁺ T cells can also become GZMB-positive and cytotoxic. Following the reviewer’s suggestion, we checked previous studies for this finding. Unfortunately, the suggested manuscript by Maehara *et al.* did not include scRNA-seq data.

In a manuscript by Lu *et al.* (PMID: 37561593), the authors analyzed blood cells from patients with IgG4-related disease using scRNA-seq. As the reviewer stated, CD4⁺ CTLs were observed. Notably, the data also revealed ISG-T cells characterized by the expression of interferon-stimulated genes such as *IFI6*, *IFI44L*, *XAF1*, *EIA2AK2*, and *OAS3* (as observed in ISG-T cells in our analysis in ANCA and SLE datasets). These results suggest that in diseases where CD4⁺ CTLs are expanded (e.g., IgG4-RD), CD4⁺ T cells are stimulated by IFN-I, leading to differentiation into CD4⁺ CTLs. We included this point in the Discussion section. The figures are shown only for the reviewers because, with minor adjustments, they are from the manuscript by Lu *et al.*

[FIGURE REDACTED]

CD4⁺ ISG-T cells exist in the blood of patients with IgG4-RD. **a** UMAP showing different clusters of T cells isolated from the blood of IgG4-RD patients. **b** Violin plots showing marker genes in each cluster. Note that the ISG-T cell cluster is positive for CD4 but not CD8. **c** Heatmap showing the expression of detailed marker genes. Interferon-stimulated genes are highly expressed in the CD4⁺ ISG-T cell cluster. With minor adjustments, these figures are from the Lu *et al.* manuscript.

4. The authors are very vague about the function and role of these new cell population(s).

We apologize for the insufficient explanation regarding the ISG-T cell function and role. We propose that ISG-T cells emerge to efficiently expand the pool of cytotoxic T lymphocytes (CTLs).

Our findings indicate that: (1) IFN-I induces ISGs in T cells (Supplementary Fig. 2); (2) ISG-T cells form a distinct cluster following unbiased clustering (Fig. 1, 2, 3), suggesting a unique T cell differentiation state; (3) ISG-T cells are in an early differentiation state, as supported by surface marker analysis (Supplementary Fig. 3) and pseudotime analysis (Fig. 3a-f); (4) T cell activation by IFN-I ultimately drives T cell differentiation into cytotoxic lymphocytes (Fig. 3g, h); (5) This IFN-I-mediated differentiation differs from the typical IL-12-mediated T cell differentiation because IL-12 induces only IFN- γ production without GzmB production (Supplementary Fig. 6g, h).

Based on these observations, we hypothesize that in diseases where IFN-I is produced, ISG-T cells emerge to differentiate into CTLs (but not IFN- γ -producing T cells). This point was added to the Discussion section (please see page 15).

5. I do not entirely understand why the authors claim that these cells develop in the kidney, while they find them in the peripheral blood of the patients? No data from peripheral blood of nephritic mice is shown.

In response to the reviewer's comment, we examined the levels of IRF7, a marker of IFN-I signaling, in CD4⁺ and CD8⁺ T cells from the blood, spleen, and kidney of nephritic mice at day 10 of the cGN model. Our results show that IRF7 levels were significantly elevated in kidney T cells compared to those in the blood and spleen (this data is now included in the revised Fig. 6e). Moreover, GzmB production was detected only in kidney T cells, but not in blood or spleen T cells (please see our response to comment 7 below). These data suggest that T cells in the kidney are exposed to higher levels of IFN-I, which is required for the induction of cytotoxicity and the development of ISG-T cells.

IRF7 expression in T cells isolated from different organs. IRF7 levels in T cells were analyzed by flow cytometry. T cells were isolated from the kidney, blood, and spleen 10 days after cGN induction.

In addition, as detailed in our response to comment 2, our single-cell spatial transcriptomics analysis identified the presence of ISG-T cells in close proximity to pDCs, the primary producers of IFN-I. These findings suggest that ISG-T cells were, at least in part, activated locally within inflamed kidney tissues.

Regarding the detection of ISG-T cells in patient blood, we speculate that some ISG-T cells migrate from inflamed kidney tissues into circulation. Tissue-specific T cells, such as follicular helper T cells, which usually reside in lymphoid organs, can also be detected in the blood because some of them emigrate from lymphoid tissues (PMID: 35277664, PMID: 33199863, PMID: 21215658). Likewise, we speculate that ISG-T cells can emigrate from inflamed tissues.

6. What time point after induction of nephritis has been evaluated in Figure 6? It would help to also show peripheral blood data.

We apologize for the lack of clarity regarding the time point. In Figure 6, the cells were analyzed ten days after glomerulonephritis induction. This information is now included in the figure legend.

As suggested by the reviewer, we added new scRNA-seq data from blood cells. Consistent with the findings in spleen data, we did not observe the expression of IFN-I genes (*Ifna1*, *Ifnb1*) in blood leukocytes. These new results are now included in the revised Figure 6d.

Blood leukocytes do not express IFN-I genes. **a** UMAP plot showing different clusters of leukocytes isolated from the blood of nephritic mice. Cells were analyzed 10 days after cGN induction. **b** UMAP plot showing *Ifnb1* expression in blood leukocytes.

7. It would help to show CD4+GzmB+ and CD8+GzmB+ T cell dynamics during the mouse model in various organs (sec lymphoid organs and kidney).

As recommended by the reviewer, we analyzed the dynamics of CD4⁺ GzmB⁺ and CD8⁺ GzmB⁺ T cells in the spleen, blood, and kidney at days 0, 3, 10, and 20 of the cGN model. Our results show that, while T cells in the blood and spleen did not exhibit increased GzmB production over time, T cells in the kidney displayed elevated GzmB production as early as day 3, peaking at day 10. These findings are shown in the revised Figure 6f.

Time kinetics analysis of GzmB production in T cells across organs. GzmB production in CD8⁺ and CD4⁺ T cells was analyzed by flow cytometry on days 0, 3, 10, and 20 after cGN induction. T cells were isolated from the kidney, blood, and spleen.

8. Anti-IFN γ treatment has been started together with the induction of nephritis. Why was this setting used and how was the dose evaluated? This treatment is rather unspecific to really test the hypothesis. Data showing immune cell infiltration into the kidney is sparse. How are T cells, neutrophils and macrophages infiltrating the kidney? Why is there no difference in CD4+GzmB+ T cells and in albuminuria (Supplemental Figure)?

Based on the expectation that IFN-I would be produced from the early phase of the disease model onward, we started neutralization concurrently with nephritis induction. The antibody dose was decided on in line with previous publications (PMID: 24789914, PMID: 32101732). In response to the comment, we quantified the numbers of T cells, neutrophils, and macrophages using immunohistochemical staining. The infiltration of T cells, neutrophils, and macrophages was significantly reduced in the treatment group, potentially reflecting milder tissue inflammation. These data are included in the revised supplementary Fig. 7 g-j.

Immune cell infiltration was reduced in the anti-IFNAR1 Ab treatment group. The numbers of T cells in glomeruli (a) and in tubulointerstitial areas (b), neutrophils (c), and macrophages (d) were quantified by immunohistochemistry.

In our murine cGN model, we did not observe many CD4 CTLs, potentially limiting our ability to analyze the IFN-I neutralization effect. Regarding albuminuria, the lack of a statistically significant difference could be due to the relatively small number of mice in each group.

As pointed out by the reviewer, the neutralization experiment was not very specific. To investigate the role of IFN-I specifically in T cells during glomerulonephritis, we obtained *Ifnar1* knockout mice and littermate WT mice. We isolated T cells from the spleen of these mice and transferred them into *Rag1* knockout mice, in which lymphocytes do not exist. In this model, T cells are deficient in IFNAR1, but other cells are wildtype. We induced a crescentic glomerulonephritis (cGN) model two weeks after the T-cell transfer. Ten days following cGN induction, the mice with IFNAR1 knockout T cells showed a significantly milder disease phenotype. Moreover, *Ifnar1* KO T cells displayed significantly lower IRF7, GzmB, and surface CD107a levels. Taken together, we propose that IFN-I plays a role in CTL development and progression of glomerulonephritis. These data are shown in the revised Fig. 5a-f and Supplementary Fig. 7a-d.

***Ifnar1* deficiency in T cells attenuates cGN in mice.** **a**, CD4⁺ and CD8⁺ T cells were isolated from the spleen of *Ifnar1* KO or wildtype mice and transferred into *Rag1* KO mice. **b**, Representative photographs of PAS staining of the kidney. Quantification is also shown. **c** and **d**, Bar graphs displaying BUN and albuminuria levels. **e** and **f**, Representative histograms and bar graphs showing the MFI of IRF7 in CD8⁺ T cells (**e**) and CD4⁺ T cells (**f**) isolated from the kidney. **g** and **h**, Representative flow cytometry plots and bar graphs showing the production of GzmB in CD8⁺ T cells (**g**) and CD4⁺ T cells (**h**) isolated from the kidney. **i** and **j**, Bar graphs displaying the surface CD107a expression on CD8⁺ T cells (**i**) and CD4⁺ T cells (**j**) isolated from the kidney. P values were calculated using the Mann-Whitney test. Data are mean + S.E.M (* $p < 0.05$, ** $p < 0.01$, *** $p < 0.001$).

Reviewer: 2

Remarks to the Author:

The study highlights the role for IFN-I in driving T cell cytotoxicity and tissue damage in ANCA-associated vasculitis (AAV) and systemic lupus erythematosus (SLE), the mechanisms underlying T cell activation remain inadequately explained. There are concerns about deficiencies in the present work that limit enthusiasm for publications.

We thank the reviewer for the productive comments.

1. Throughout the article, there appear to be several instances of over-interpretation of the research findings, leading to conclusions that may seem somewhat speculative. For example, in line 95, there is a lack of data analysis or experimental evidence supporting the activation and differentiation of T cells stimulated by type I interferon into ISG-T/ISG-CTL. Moreover, there is no analytical data provided to confirm whether ISG-CTL represents an intermediate phenotype between ISG-T and CTL.

We sincerely appreciate the valuable comments and suggestions. We performed the additional analyses discussed below.

A. Data analysis or experimental evidence supporting the activation and differentiation of T cells stimulated by type I interferon into ISG-T/ISG-CTL

In response to the reviewer's comment, we investigated whether stimulation by type I interferon (IFN-I) induces the expression of interferon-stimulated genes (ISGs), a key transcriptomic feature of ISG-T cells.

First, we analyzed human T cell transcriptome data (GSE17302, PMID: 21108462). In this experiment, CD8⁺ CD45RO⁻ T cells from 3 healthy donors were stimulated with IFN- α *in vitro* for 48 hours. As shown below, T cells upregulated numerous interferon-stimulated genes, such as *ISG15*, *ISG20*, *IFIT1*, and *IRF7*. These findings demonstrate that human T cells can respond to type I interferon and substantially upregulate ISGs to differentiate into ISG-T cells.

IFN-I induces ISGs in human T cells. Heatmap showing the ISG expression in human CD8⁺ T cells. T cells were cultured *in vitro* in the presence of IFN- α for 48 hours and analyzed by microarray.

Next, we investigated the effect of type I interferon in vivo by analyzing murine T-cell transcriptome data (GSE75202, PMID: 26824662). In this experiment, mice were subcutaneously injected with IFN- α , and spleen T cells were analyzed 2 hours after injection. Both CD4⁺ and CD8⁺ T cells showed significantly higher expression of ISGs, including *Isg15*, *Isg20*, *Ifit3*, *Ifit1*, and *Irf7*, compared to the control group. These findings demonstrate that IFN-I induces ISG expression in T cells in vivo.

IFN-I induces ISGs in murine T cells. Heatmap showing the expression of ISGs in murine CD4⁺ and CD8⁺ T cells. Spleen T cells were analyzed using microarray 2 hours after subcutaneous IFN- α injection.

Together, these results demonstrate that IFN-I induces the ISG expression, a unique, defining feature of ISG-T cells observed as a distinct cluster in our scRNA-seq analyses. These results are shown in the revised text and Supplementary Fig. 2.

B. Analytical data to confirm whether ISG-CTL represents an intermediate phenotype between ISG-T and CTL.

We sincerely thank the reviewer for this insightful comment. Accordingly, we deleted the sentence “exhibiting an intermediate phenotype” from the manuscript.

To infer the differentiation state of the ISG-CTL cluster, we performed pseudotime analysis. The results revealed that the ISG-CTL cluster exhibits a higher pseudotime score compared to the ISG-T cell cluster, suggesting that the ISG-CTL cluster represents a more mature population. However, both the ISG-CTL and CTL clusters showed comparably high pseudotime scores. Based on these findings, along with other experimental results showing that IFN-I induces cytotoxicity in T cells, we hypothesized in the Discussion section that ISG-T cells differentiate into the ISG-CTL population. The results are presented in the revised Supplementary Fig. 11b.

Pseudotime analysis of CD4⁺ and CD8⁺ T cells in the SLE dataset. Pseudotime scores were calculated using Monocle 3, with CD4⁺ and CD8⁺ Tcm/Tn clusters designated as the root of differentiation. While ISG-T cells exhibited relatively lower pseudotime scores, ISG-CTLs and CTLs showed comparably high pseudotime scores.

2. The expression patterns of marker genes used to define various cell types in this study appear unclear. For instance, Fig 1C and 2E indicate no notable expression of highly specific genes in CD4+TEM. Furthermore, the CTL cells in the ANCA dataset do not exhibit significant expression levels of CD4 and CD8A, whereas CTL cells in the Lupus dataset show high expression of CD8A. Could you clarify if these two cell types are considered identical? Additionally, what is the rationale for separately analyzing the ANCA dataset from the Lupus dataset?

We apologize for the lack of clarity regarding the marker genes. In the revised heatmap, we additionally included *S1PR1*, *CD44*, and *KLF2*. CD4+ Tem cells are defined as *CD4+ CD44+ SELL- CCR7- S1PR1+ KLF2+* cells. These marker genes were selected based on previous studies (PMID: 34914499, PMID: 32769171, PMID: 28930685, PMID: 27776108, PMID: 38187377). This information is given in the Methods section.

Marker genes of different clusters in ANCA and Lupus datasets. Heatmap showing the expression of marker genes in each cluster.

The ANCA and SLE datasets were derived from different patient cohorts using different sample preparation protocols. To minimize batch effects, we separately analyzed these datasets. Given the differences in disease pathogenesis, we also acknowledge that CTL cells in the ANCA and SLE datasets may exhibit an inherent difference. However, for simplicity and due to their similar transcriptomic features (e.g., cytotoxic gene expression), we annotated these clusters as CTL in both datasets.

3. The number of patient samples exceeds that of healthy donors (HD). Are the patient cell counts also significantly higher than those of HD? If so, how can we determine that ISG-T and other cells have expanded based on that ANCA or SLE cells make up more than 50% of the total? Could statistical analysis be included to substantiate the expansion of ISG-T and other cells? Additionally, could the ISG score in Figure 1D be presented with statistical analysis?

We analyzed cell proportions (percentages) rather than absolute cell counts to compare across different individuals and conditions.

In response to the reviewer’s suggestion, we statistically analyzed the proportion of ISG-T cells. While the increase in the proportion of ISG-T cells in ANCA-associated vasculitis patients did not reach statistical significance ($p = 0.08$), both ISG-T cells and ISG-CTLs were significantly increased in SLE patients. These results are shown in the revised Supplementary Fig. 1i and j.

ISG-T cell percentage increases in AAV or SLE patients. **a** Bar graph showing the fraction of ISG-T cells in the blood. **b** Bar graphs showing the percentage of ISG-T cells (left) and ISG-CTLs (right) among blood T cells.

In addition, we performed a statistical analysis (Wilcoxon rank-sum test) on the ISG score, as suggested. The test result ($p < 0.001$) was added to the revised Fig. 1d.

4. The Figure legend throughout the article do not specify the statistical analysis methods corresponding to the figures.

We apologize for omitting the inclusion of statistical analysis details in the figure legends. The relevant statistical analysis information is now in all figure legends throughout the article.

5. As depicted in Figure 2E, GZMB and GZMA exhibit higher expression levels in CD8 TEM cells. Meanwhile, Figure 2F indicates that GZMB and GZMK are not predominantly expressed in ISG-T cells, Figure 2G shows that only a small subset of cells with elevated ISG scores also demonstrate high cytotoxicity scores. Thus, further investigation is required to determine whether ISG-T cells exhibit a cytotoxic phenotype. The co-expression of ISG genes and cytotoxicity-related genes can be visualized in these data, and multi-color tissue immunofluorescence can be utilized to verify the co-expression of ISG and cytotoxicity-related genes in kidney tissues of the cGN mouse model.

As stated, cytotoxicity-related genes are expressed in both ISG-T cells and other CD8 T-cell clusters. We speculate this observation is because IFN-I signaling is not the sole pathway for CTL differentiation or is not required to maintain a cytotoxic phenotype.

In response to the reviewer’s comment, we analyzed the co-expression of ISG and cytotoxicity scores within the ISG-T cell cluster. Our analysis revealed a positive correlation between the two scores, with a *p*-value of 0.0081. This new data was added to Supplementary Fig. 4h.

The ISG score is positively correlated with the cytotoxicity score in ISG-T cells. Scatter plot showing the correlation between ISG and cytotoxicity scores in the ISG-T cell cluster (scRNA-seq data from the murine cGN model). The *p*-value for the correlation was calculated using Pearson correlation.

Moreover, to verify the expression of GzmB in ISG-T cells, we isolated T cells from the kidneys and analyzed GzmB production in CD8⁺ IRF7⁺ ISG-T cells. For comparison, we analyzed CD8⁺ IRF7⁻ CD44⁻ T cells as a control group. Our analysis confirmed that IRF7⁺ ISG-T cells express significantly higher GzmB levels than immature IRF7⁻ CD44⁻ CD8⁺ T cells. These results are shown in the revised Fig. 4i and j.

ISG-T cells show GzmB production in a murine cGN model. T cells were isolated from kidneys on day 10 of the cGN model mice and analyzed for IRF7 and GzmB expression. Higher levels of GzmB production were observed in IRF7⁺ ISG T cells compared to IRF7⁻ CD44⁻ cells in both CD8⁺ T cells (a) and CD4⁺ T cells (b).

6. As is widely recognized, Monocle3 can designate the starting point of differentiation trajectories. The authors specified naïve T cells as the starting point. Was the starting position of ISG-T manually specified? The research method does not clarify the basis for calculating the pseudotime score. Besides Monocle3, there are numerous other methods such as RNA velocity analysis capable of conducting pseudotime trajectory analysis. Can these alternative methods lead to similar conclusions?

Our Monocle3 analysis designated only naïve T cells as the starting point of differentiation trajectories. Without manually setting ISG-T cells as the starting point, these cells exhibited a low pseudotime score. We apologize for the omission and described the pseudotime analysis in the Methods section. In the revised text, we clarified that we only designated naïve T cells as the root of differentiation in the pseudotime analysis using Monocle 3.

As suggested by the reviewer, we performed RNA velocity analysis using scVelo. We exclusively focused on the cGN datasets for this analysis because RNA splicing conditions may differ significantly between healthy control and nephritic conditions, even within the same cluster. RNA velocity analysis does not require setting a specific cluster as a root (as opposed to Monocle3). As shown below, we identified an ISG-T cell cluster characterized by the expression of *Isg15* and *Irf7*. Consistent with our pseudotime analysis using Monocle3, the Velocity_Pseudotime analysis also demonstrated that the pseudotime score was significantly lower in ISG-T cells than in other CD8⁺ T cell clusters. In this cGN dataset, the pseudotime score was lowest in the ISG-T cell cluster because not many naïve T cells exist in the nephritic kidney (we did not identify distinct naïve T cell clusters in the cGN dataset). These findings support our conclusion that ISG-T cells exhibit a lower pseudotime score, suggesting a potentially immature phenotype. (Figures are shown only for reviewers.)

ISG-T cells show a low pseudotime score in RNA velocity analysis. **a** RNA velocity plot computed by scVelo. **b** UMAP and violin plots showing the Velocity_Pseudotime score across clusters (**** P < 0.0001, Wilcoxon rank-sum test). **c** UMAP plots illustrating the *Isg15*, *Irf7*, *Gzmb*, and *Mki67* expression levels.

7. In addition to GZMB, there are many other indicators for detecting cytotoxicity levels, such as the expression of CD107a, etc. The expression level of GZMB alone cannot lead to the conclusion that IFN-I drives T cell cytotoxicity (line 156).

Thank you for this insightful comment. In response to the reviewer’s suggestion, we investigated surface CD107a as a marker of degranulation, both in vivo and in vitro, using wildtype T cells and *Ifnar1* knockout T cells.

First, T cells stimulated with IFN-β1 in vitro showed significantly higher levels of surface CD107a, indicating that IFN-β1 not only promoted GzmB production but also increased cytotoxic activity. These results are included in the revised Supplementary Fig. 6d.

IFN-β1 increases surface CD107a in both CD4+ and CD8+ T cells. T cells were stimulated with CD3/CD28 antibodies in the presence or absence of IFN-β1 for 24 hours, followed by the analysis of surface CD107a levels as a marker of degranulation.

Next, to further validate the role of IFN-I in the induction of cytotoxicity, we conducted *in-vitro* experiments using *Ifnar1* KO T cells. In comparison to wildtype T cells, *Ifnar1* KO T cells exhibited significantly lower levels of GzmB production, surface CD107a expression, and IRF7 expression. These findings underscore that IFN-β1 induces cytotoxicity and T cell degranulation primarily through IFNAR1-dependent signaling and are included in the revised Supplementary Fig. 6e,f,i. For details on *in-vivo* experiments using *Ifnar1* KO T cells, please see our response to comment 9.

IFN-β1 does not induce cytotoxicity in *Ifnar1* KO T cells. WT and *Ifnar1* KO T cells were stimulated with CD3/CD28 antibodies and IFN-β1, followed by the analysis of IRF7 expression, GzmB production, and surface CD107a levels. GzmB production in CD4+ T cells was analyzed two days after the stimulation. For other analysis, cells were stimulated for 24 hours.

8. As depicted in Figure 4B, STAT1 exhibits high average expression in ISG-T, and also plays a crucial role in T cell differentiation. Has there been an investigation into whether the expression level of STAT1 changes after IFN-I neutralization, and whether STAT1 expression correlates with GZMB expression? Additionally, how does IRF7 regulate the levels of GZMB?

Stat1 expression levels after IFN-I stimulation

It is well established that STAT1 plays a key role in IFN-I signaling (PMID: 25614319). Although we did not investigate STAT1 expression levels in the IFN-I neutralization experiment, we analyzed T cell transcriptome datasets (GSE75202, PMID: 26824662), observing an increase in *Stat1* mRNA levels in IFN-I-treated T cells. (This result is shown only for the reviewers.)

IFN-I induces *Stat1* expression.

Bar graph showing the expression of *Stat1* in murine T cells. Spleen T cells were analyzed with microarray 2 hours after subcutaneous IFN-α injection.

Correlation between *Stat1* and *Gzmb*

To analyze the correlation between *Stat1* and cytotoxicity-associated genes, we performed an additional analysis of our scRNA-seq data from the murine cGN model (as previously done in response to comment 5). *Gzmb* expression was detected in only a fraction of ISG-T cell clusters. Therefore, we did not observe a significant correlation between *Stat1* and *Gzmb* expression. However, *Stat1* exhibited a significantly positive correlation with the cytotoxicity score, which includes genes such as *Gzma*, *Gzmb*, *Gzmk*, and *Prf1*. These findings suggest that *Stat1* is associated with the induction of cytotoxicity in T cells. (Figures are shown only for the reviewers.)

Stat1 shows a positive correlation with the cytotoxicity score in ISG-T cells.

Scatter plots showing the correlation between *Stat1* and *Gzmb* expression (left) and the correlation between *Stat1* and the cytotoxicity score (right).

Regulatory mechanism of IRF7 on Gzmb

We observed an increase in *Stat1* expression in IRF7-overexpressing T cells, suggesting that IRF7 may, at least in part, contribute to Gzmb production by regulating Stat1. This data is now included in the revised Supplementary Fig. 6j. While we agree that a detailed investigation into IRF7's regulatory mechanisms for Gzmb would be valuable, we consider it beyond the scope of the present manuscript.

IRF7 overexpression induces *Stat1* mRNA expression.

Bar graph showing the expression of *Stat1*. In the control group, dominant-negative IRF7 was overexpressed in T cells. In the IRF7 group, constitutively active IRF7 was transduced. Cells were analyzed one day after transduction.

9. In lupus nephritis or cGN mouse model, ISGs are highly expressed not only in CD4 and CD8 T cells, and the inhibition of IFNAR1 antibody at disease onset may not solely affect ISG-T. Modeling CD4- and CD8-specific knockout of IRF7 or IFNAR receptors would provide clearer insights into how IFN-I influences disease progression through regulation of ISG-T.

As pointed out, IFN-I affects different cell types, and the neutralization experiment was not specific to T cells. To investigate specifically the role of IFN-I in T cells during glomerulonephritis, we conducted experiments using *Ifnar1* knockout mice and littermate WT controls. T cells were isolated from the spleens of these mice and transferred into *Rag1* knockout mice, which are deficient in lymphocytes. In this model, T cells are deficient in IFNAR1, but other cells are wildtype. Two weeks after T cell transfer, we induced a crescentic glomerulonephritis (cGN) model. Ten days after cGN induction, mice with *Ifnar1* knockout T cells exhibited a significantly milder disease phenotype. Furthermore, *Ifnar1* KO T cells showed markedly reduced IRF7, GzmB, and CD107a levels. These findings support the conclusion that IFN-I plays a critical role in cytotoxic-T-lymphocyte (CTL) development and the progression of glomerulonephritis. These data are shown in the revised Fig. 5a-f and Supplementary Fig. 7a-d.

***Ifnar1* deficiency in T cells attenuates cGN in mice.** **a**, CD4⁺ and CD8⁺ T cells were isolated from the spleen of *Ifnar1* KO or wildtype mice and transferred into *Rag1* KO mice. **b**, Representative photographs of PAS-stained kidney sections. Quantification is also shown. **c** and **d**, Bar graphs showing BUN (**c**) and albuminuria (**d**) levels. **e** and **f**, Representative histograms and bar graphs displaying the MFI of IRF7 in CD8⁺ T cells (**e**) and CD4⁺ T cells (**f**) isolated from the kidney. **g** and **h**, Representative flow cytometry plots and bar graphs showing the production of GzmB in CD8⁺ T cells (**g**) and CD4⁺ T cells (**h**) isolated from the kidney. **i** and **j**, Bar graphs displaying the surface CD107a expression on CD8⁺ T cells (**i**) and CD4⁺ T cells (**j**) isolated from the kidney. P values were calculated using the Mann-Whitney test. Data are mean + S.E.M (* $p < 0.05$, ** $p < 0.01$, *** $p < 0.001$).

10. Is IRF7 elevated in infection models, and then is there a regulatory report that increases GZMB expression?

We did not find published studies reporting *IRF7* expression levels in T cells from infection models or patients with infection. To address the reviewer's question, we analyzed a publicly available scRNA-seq dataset of blood T cells from patients with SARS-CoV-2 infection and healthy donors (GSE163668). While *IRF7* expression was elevated across different T-cell populations in the patient group, ISG-T cells specifically showed a marked increase in *IRF7* expression levels. Furthermore, *GZMB* expression levels and cytotoxicity scores in ISG-T cells were significantly higher in the patient group than in healthy donors. These findings are presented in the revised Supplementary Fig. 12a-d.

IRF7 expression in ISG-T cells is elevated in patients with SARS-CoV2. **a** scRNA-seq data of blood T cells from patients with SARS-CoV2 and healthy donors (GSE163668) were analyzed. **b** UMAP plot showing different T cell clusters. **c** Heatmap displaying the expression of marker genes in each cluster. **d** Violin plots showing ISG score, *IRF7* expression, cytotoxicity score, and *GZMB* expression levels in the SARS-CoV2 and healthy donor groups.

In a study by Zhou et al. (PMID: 22875973) using *Irf7* knockout (KO) mice, virus-specific CD8 T cells failed to expand during lymphocytic choriomeningitis virus infection. However, to our knowledge, no other studies directly investigated the regulatory role of IRF7 in *GZMB* expression.

11. In the spatial transcriptome data of ANCA-GN patients, do the regions with high interferon scores and cytotoxicity scores correspond to CD4 or CD8 T cells? Do these regions exhibit high expression of the *IRF7* gene?

In the sequencing-based spatial transcriptome analysis (Visium), the resolution is not at the single-cell level, making it challenging to conclude definitively whether ISGs and cytotoxicity-associated genes are specifically expressed by T cells. However, in response to the reviewer's comment, we analyzed the spatial transcriptome data using a deconvolution technique, enabling us to estimate the T-cell frequency in different regions. This analysis revealed a significant enrichment of T cells in inflamed areas, suggesting that higher ISG and cytotoxicity scores may reflect T cell enrichment within these inflamed regions.

In addition, *IRF7* expression was significantly higher in inflamed periglomerular and tubulointerstitial areas than in healthy counterparts. These results are now included in the revised Supplementary Fig. 9e, f.

T cell abundance and *IRF7* expression levels are elevated in inflamed tissue compartments in ANCA-GN. a Graphs showing T cell abundance in inflamed and healthy regions based on deconvolution analysis. **b** Graphs displaying *IRF7* expression in inflamed and healthy regions.

To further address the low-resolution limitations of the Visium, we established an imaging-based single-cell transcriptomics technique using the Xenium platform. While genes encoding IFN-I and cytotoxic molecules were not sensitively detected with this system, we successfully identified ISG-T cells in kidney tissues through unsupervised clustering and annotation based on the expression of T cell marker genes and ISGs. Notably, cell proximity analysis revealed the presence of ISG-T cells in close proximity to plasmacytoid dendritic cells (pDCs), the major producers of IFN-I, among non-lymphocyte cell types. We included these data in revised Fig. 9 and Supplementary Fig. 10. (Main data are also shown in the next page.)

Imaging-based single-cell spatial transcriptomics reveals the presence of ISG-T cells in close proximity to plasmacytoid dendritic cells in inflamed kidneys. **a** Kidney tissue was analyzed using imaging-based single-cell spatial transcriptomics. **b** UMAP plot showing different cell clusters. **c-e** Representative images of kidney tissues under different conditions: H&E-stained tissue overview (left), magnified view of the indicated area (upper middle), DAPI-stained image overlaid with cell boundaries and specific marker genes (upper right), segmented cells color-coded by cell type annotation using the same color scheme as in **b** (lower middle), and segmented cells showing only ISG-T cells and pDCs in color with other cells shown in transparent (lower right). **f** Heatmap showing ISG expression levels in different T cell subtypes. **g** Dot plot showing the frequency of ISG-T cells among all T cells. **h** Cell proximity analysis showing the median distances between ISG-T cells and different non-lymphocyte cell types. **i** Bar graph showing the distances between ISG-T cells and pDCs in different conditions. Error bars represent the standard deviation. P values were calculated by one-way ANOVA with Tukey's multiple comparison test (* $p < 0.05$, **** $p < 0.0001$). For cell type abbreviations, please see figure legend for Fig. 9

12. There is potential for improvement in thesis writing, particularly with managing English tenses.

Thank you for the comment. In response, we had the manuscript professionally reviewed and edited by a native English speaker through a professional editing service.

Reviewer: 3

Remarks to the Author:

This manuscript explores the ability of type I interferons (IFNs) to drive T cell cytotoxicity and subsequent renal damage within ANCA associated vasculitis (AAV) and systemic lupus erythematosus (SLE), systemic autoimmune diseases that involve glomerulonephritis. Using RNAseq data, this study has identified a unique subset of T cells characterised by the expression of type I IFN genes, ISG-T cells, that are expanded in AAV and SLE. They show that these subsets are present in mouse models of crescentic glomerulonephritis (cGN) and are associated with cytotoxicity through IRF7. They also show a correlation between interferon-stimulated gene (ISG) expression and genes related to cytotoxicity in the ISG-T subsets from SLE and AAV human kidney samples. This work offers novel insights into the role of type I IFN mediated T cell toxicity in autoimmune diseases involving renal inflammation and I commend the authors for their thorough and interesting work.

We thank the reviewer for the very supportive, constructive, and insightful comments.

Comments:

1. A detailed participant table for all human samples is required for appropriate interpretation of the data provided. Factors such as age, sex, medication, disease activity/severity at the time of sampling and co-morbidities, can all have dramatic effects on gene expression. Particular factors, such as age and sex, may need to be included in your statistical models to ensure that the changes that you describe are disease specific. This is required for data presented in Figures 1, 7 and 8.

In response to the reviewer's comment, we produced participant tables for all datasets, including key parameters such as age and sex. Additional information, such as kidney function, treatment, and disease activity, was included where applicable. These tables are provided as a supplementary Excel file.

2. A major issue is that the authors have not really proven mechanism with regard: (i) that in patients or their mouse model, that IRF7 is the definitive transcription factor needed for ISG-T cell development and (ii) a mechanism for how IRF7 is regulating granzyme B (or A or K) expression in patients and/or their model. Point (i) would probably require knock out animals and thus, likely beyond the scope of this work, but I think it is very important that the authors word their interpretation of what they have done and highlight the limitations/alternatives. For point (ii), this could be explored further. A possibility could be the IRF7 is promoting/amplifying further IFN-I responses, which in T cells, could be working via IFN-I driven STAT5 activation to drive GZMB expression.

We thank the reviewer for the productive comments.

i) Further analysis using knockout mice

This comment is much appreciated. To underscore the role of the type I interferon – IRF7 pathway, we tried to obtain both *Ifnar1* KO mice and *Irf7* KO mice. However, only *Ifnar1* KO mice were available. Therefore, to investigate the role of IFN-I specifically in T cells during glomerulonephritis, we used *Ifnar1* knockout mice and littermate WT controls. We isolated T cells from the spleen of these mice and transferred them into Rag1 knockout mice, in which

lymphocytes do not exist. In this model, T cells are deficient in IFNAR1, but other cells are wildtype. Two weeks after the T-cell transfer, we induced a crescentic glomerulonephritis (cGN) model. Mice with IFNAR1 knockout T cells showed a significantly milder disease ten days after cGN induction. Moreover, *Ifnar1* KO T cells exhibited significantly lower IRF7, GzmB, and CD107a levels. Taken together, we consider that IFN-I plays a key role in CTL development and progression of glomerulonephritis. These data are shown in the revised Fig. 5a-f and Supplementary Fig. 7a-d. Since the focus of these experiments was on IFN-I signaling, not IRF7 signaling, this limitation of our study is discussed on page 17.

***Ifnar1* KO in T cells attenuates the progression of cGN in mice.** **a**, CD4⁺ and CD8⁺ T cells were isolated from the spleen of *Ifnar1* KO or wildtype mice and transferred into Rag1 KO mice. **b**, Representative photographs of PAS-stained kidney sections. Quantification is also shown. **c** and **d**, Bar graphs showing BUN (**c**) and albuminuria (**d**) levels. **e** and **f**, Representative histograms and bar graphs displaying the MFI of IRF7 in CD8⁺ T cells (**e**) and CD4⁺ T cells (**f**) isolated from the kidney. **g** and **h**, Representative flow cytometry plots and bar graphs showing the production of GzmB in CD8⁺ T cells (**g**) and CD4⁺ T cells (**h**) isolated from the kidney. **i** and **j**, Bar graphs displaying the surface CD107a expression on CD8⁺ T cells (**i**) and CD4⁺ T cells (**j**) isolated from the kidney. P values were calculated using the Mann-Whitney test. Data are mean + S.E.M (* $p < 0.05$, ** $p < 0.01$, *** $p < 0.001$).

ii) Mechanism of how IRF7 regulates granzyme B

As recommended by the reviewer, we analyzed the expression of *Stat1* and *Stat5*, which are known to play roles in T-cell differentiation and cytotoxicity in IRF7-transduced T cells. While *Stat5* levels did not increase, *Stat1* levels were elevated upon IRF7 overexpression. The *Stat1* data are included in the revised Supplementary Fig. 6j.

IRF7 induces the expression of *Stat1*. Bar graphs showing the expression of *Stat1*, *Stat5a*, and *Stat5b*. T cells were retrovirally transduced with the control gene (dominant-negative IRF7) or constitutive active IRF7 and analyzed by qPCR one day after transduction.

3. The “Lupus Nephritis” patients analysed in Figure 7, did they have cGN as their kidney pathology?

We appreciate your interest in understanding the specific kidney pathology of the lupus nephritis patients presented in Figure 7. Unfortunately, we were unable to access detailed pathological information from the database.

4. Can you clarify heat maps in Figure 1 C whether these are comparing disease samples to HD or whether these compare between T cells subtypes? Can you also make the figure scales the same for the ANCA data set and SLE data set. How were the marker genes shown in Fig 1C chosen?

We apologize for the confusing explanation. The heatmaps in Fig.1c compare different T-cell subsets (but not HD vs. disease). The figure legend was revised accordingly.

In response to the reviewer’s suggestion, we adjusted the scales of the figures in the revised Figure 1c to be the same for both datasets.

The marker genes were selected based on previous publications (PMID: 34914499, PMID: 32769171, PMID: 28930685, PMID: 27776108, PMID: 38187377). This information is included in the Methods section.

5. Page 5, line 90-91 “The ISG-T cell population expanded in the blood of patients versus healthy donors, suggesting an IFN-I mediated activation of T cells in autoimmune disease” – this is clear from the graphs in SLE but less so in AAV. Similarly, on page 6 line 123-124 the differences are unclear. Can you include percentages in the text? I would also suggest including a section in the discussion regarding the consistently lower IFN expression within AAV samples compared to SLE (Figure 1 and Figure 7) with respect to current literature in the field.

The percentages are now given in the text.

As the reviewer pointed out, the ISG-T cell expansion and tissue ISG signature are more pronounced in SLE/lupus nephritis patients compared to AAV/ANCA-GN patients. This likely reflects differences in IFN-I levels between these diseases, as SLE is better characterized as an IFN-I-associated disease. This point is now included in the Discussion section (please see page 16).

6. What were the n numbers for mice used in the in vivo studies?

We used ten mice in the intervention group and nine in the control group. The figure legend includes this information.

7. It is very reassuring to see the validation of the results from Figure 2 from the cGN (nephrotoxic nephritis model) mouse model in an additional model more relevant for lupus nephritis (the MRL/lpr lupus prone nephritic model results in Supp Fig 3) and really helps support the conclusions of the authors; can you do similar on a relevant ANCA-related kidney disease mouse model dataset?

We agree that including an ANCA-GN model would strengthen our manuscript. However, we did not generate an ANCA-GN model. We also searched for public datasets, but no scRNA-seq data of ANCA-GN models were publicly available.

8. In Figure 5, how similar to health control mice are the measures following administration of anti-IFNAR1 ab?

While the administration of anti-IFNAR1 Ab significantly reduced cGN severity compared to the isotype control group, the disease severity in the treatment group remained higher than in healthy mice. In healthy mice, glomerular crescents were absent, BUN levels were approximately 25 mg/dL, and the urinary albumin-to-creatinine ratio was almost zero (PMID: 19339380, PMID: 27851911). (Figures are shown only for the reviewers.)

[FIGURE REDACTED]

BUN and albuminuria levels in the control and cGN groups. Graphs showing BUN and urinary albumin-to-creatinine ratio analyses in the healthy control and cGN groups.

9. In patient included in Figure 1, did any of these have renal disease involvement and in particular, cGN?

In the ANCA dataset, two patients had cGN. In the SLE dataset, 18 patients, including three with cGN, showed kidney involvement. These details are now provided in the patient data Excel file.

10. I would strongly suggest rethinking the wording/conclusion of lines 221–223 and associated conclusions. I think this is associated with the fact that there is not enough attention paid throughout about the fact that ultimately, AAV and SLE are quite different diseases and in relation to IFN-I dysregulation, although very clear in SLE pathology, both systemically and in the kidney, in AAV, this IFN-I phenotype is strong in kidney (pt 24 below) and murine models (<https://rupress.org/jem/article/219/10/e20220759/213416/Monocyte-derived-macrophages-aggravate-pulmonary>), but less conclusive systemically, particularly studies that use properly age/sex matched healthy and disease cohorts, with appropriate positive and negative disease controls (<https://nature.com/articles/s41598-021-87760-4>).

Thank you for your insightful comments and suggestions on the differences in IFN-I dysregulation between SLE and AAV, particularly on the conclusions drawn in lines 221–223 of our manuscript. We agree that SLE and AAV are distinct diseases with different pathogenic mechanisms. This distinction is essential, especially concerning the role of IFN-I signaling.

To address your concerns, we revised the text in lines 221–223 to reflect the results analyzed accurately. Specifically, we now emphasize that the IFN-I signaling observed in our study is more pronounced in the nephritic kidney environment of ANCA-GN rather than systemically, which aligns with the findings from recent studies, including those the reviewer referred to (please see page 16).

Furthermore, we discussed the importance of considering the localized effects of IFN-I signaling in ANCA-GN and the potential differences in systemic involvement compared to SLE in the Discussion section. We also incorporated the reference you provided (please see page 16).

11. I would disagree that Supp 5 B & C support the conclusion in results text lines 184 –186 regarding improvement of proteinuria and BUN levels.

We revised the text regarding the proteinuria and BUN levels as below.

“Although albuminuria and BUN levels also decreased, these changes did not reach statistical significance.”

12. I don’t think the analysis presented in Figure 6 and Supp Fig 6 is supportive enough of the overall conclusion of that results section, that “IFN-I signaling in the inflamed kidney tissue is mediated by the cross-talk between myeloid cells and T cells”. I think the data presented is supportive of a hypothesis that is might be mediated via this mechanism, but the authors would need to present more experimental work/analysis to support that as a conclusive mechanism, so either that needs to be done or else that section toned down, particularly the section heading.

In response to the reviewer’s comment, we sought to identify the organ where T cells are activated by IFN-I. To this end, we examined IRF7, a marker of IFN-I signaling, in CD4⁺ and CD8⁺ T cells from the blood, spleen, and kidney of nephritic mice at day 10 of the cGN model. Our results showed that IRF7 levels were significantly higher in kidney T cells than in the blood and spleen. This finding suggests that T cells in the kidney are exposed to higher levels of IFN-I, which is critical for ISG-T-cell development. These results are included in the revised Fig 6e.

IRF7 levels in T cells isolated from different organs. IRF7 expression in T cells was evaluated by flow cytometry. T cells were isolated from the kidney, blood, and spleen 10 days after cGN induction.

We also analyzed the dynamics of CD4⁺ GzmB⁺ and CD8⁺ GzmB⁺ T cells in the spleen, blood, and kidney at days 0, 3, 10, and 20 of the cGN model. Our results show that, while T cells in the blood and spleen did not exhibit increased GzmB production over time, T cells in the kidney showed elevated GzmB production as early as day 3, peaking at day 10. These findings are included in the revised Figure 6f.

Time kinetics analysis of GzmB production in T cells across organs. GzmB production in CD8⁺ and CD4⁺ T cells was analyzed by flow cytometry on days 0, 3, 10, and 20 post-cGN induction. T cells were isolated from the kidney, blood, and spleen.

Finally, we conducted imaging-based single-cell spatial transcriptomics analysis using the *Xenium* platform to analyze ISG-T cells in the kidneys of patients with ANCA-GN or lupus nephritis. This analysis revealed that ISG-T cells were located in close proximity to plasmacytoid dendritic cells (pDCs), the primary producers of IFN-I, in both diseases. These new findings are included in the revised Fig. 9 and Supplementary Fig. 10.

Collectively, our findings suggest that T cells are, at least in part, locally activated in the kidney via IFN-I. However, the section heading was toned down in line with the reviewer's comment.

Imaging-based single-cell spatial transcriptomics reveals the presence of ISG-T cells in close proximity to plasmacytoid dendritic cells in inflamed kidneys. **a** Kidney tissue was analyzed using imaging-based single-cell spatial transcriptomics. **b** UMAP plot showing different cell clusters. **c-e** Representative images of kidney tissues under different conditions: H&E-stained tissue overview (left), magnified view of the indicated area (upper middle), DAPI-stained image overlaid with cell boundaries and specific marker genes (upper right), segmented cells color-coded by cell type annotation using the same color scheme as in **b** (lower middle), and segmented cells showing only ISG-T cells and pDCs in color with other cells shown in transparent (lower right). **f** Heatmap showing ISG expression levels in different T cell subtypes. **g** Dot plot showing the frequency of ISG-T cells among all T cells. **h** Cell proximity analysis showing the median distances between ISG-T cells and different non-lymphocyte cell types. **i** Bar graph showing the distances between ISG-T cells and pDCs in different conditions. Error bars represent the standard deviation. P values were calculated by one-way ANOVA with Tukey's multiple comparison test (* $p < 0.05$, **** $p < 0.0001$). For cell type abbreviations, please see figure legend for Fig. 9

13. Due to small differences in overall percentages noted with respect to GzmB, I would suggest adding the percent into the results text as the scale for these graphs can be misleading when not set to 100%.

As suggested, we included the GzmB-positive cell frequency information in the Results section text.

Transduction of T cells with constitutively active IRF7 increased GzmB expression in both CD8⁺ T cells (7.1% in GFP alone vs 9.8% in constitutively active IRF7, $p < 0.001$) and CD4⁺ T cells (0.18% in GFP alone vs 0.9% in constitutively active IRF7, $p < 0.001$) compared to controls (Fig. 4I). (page 10)

“IFNAR1 neutralization significantly reduced IRF7 levels in CD8⁺ and CD4⁺ T cells (Fig. 5i, j), decreased the frequency of GzmB-positive cells in CD8⁺ T cells (6.6% vs 3.4%, $p < 0.01$) and CD4⁺ T cells (1.8% vs 1.3%, $p = 0.10$)” (page 10-11)

14. I suggest you add Supplemental Fig. 5A to Fig. 5. (CD4 GzmB data). Although not significant, it still supports your overall message and keeps consistency across figures.

As suggested, we included the original sFig5A in the revised Fig. 5I.

15. There seems to be no gradients present in Fig. 2 G, Fig. 3 B and E, Fig. 6 G, Supplementary Fig 3. E and G, Supplementary Fig. 6. Was this on purpose? Can you make these more clear, maybe by adding multi-coloured gradients i.e. from white to orange to pink? I would also like to see consistency across the scales for the UMAPs. Some of these go from 1-3 some go from 0-2 etc. This is particularly important in Fig. 3.

Thank you for your valuable feedback. We revised the UMAPs to include gradients. Although we tried to use the suggested multi-colored gradients (white to orange to pink), this color scheme did not align well with the color codes used throughout the manuscript, nor did it improve the clarity of the visualization. Therefore, we maintained our original color scheme while ensuring the presence of better distinguishable gradients for consistency. We also standardized the scales across the UMAPs in Fig. 3 to ensure uniformity.

16. Please increase the font size of supplemental figure 1 G-H scale.

This issue of the font being too small was addressed.

17. With regards to point 1, it appears that the SLE patients analysed in Fig. 7 B may cluster into 2 separate subgroups based on the IFN-I associated genes and the authors may wish to comment on the heterogeneity of this cohort/human cohorts.

Thank you for your insightful observation regarding the clustering of SLE patients in Fig. 7B. We acknowledge that the patients appear to be two distinct subgroups based on the expression of IFN-I-associated genes. This observation likely reflects the heterogeneity within the SLE cohort, a well-documented characteristic of human SLE populations.

This heterogeneity is discussed in the manuscript, and the potential implications for understanding the diverse immune responses in SLE patients are emphasized (please see page 16).

18. Please make the scales for the SLE and ANCA data in Figure 7 C the same. The HD look different on each graph raising concerns on reliability of data though I think this is due to the scaling.

Thank you for bringing this observation to our attention. We adjusted the scales in the SLE and ANCA datasets in Figure 7 to ensure that the healthy donors appear consistent across the graphs.

19. Can you include the quantity of RNA used for PCR work in the methods section?

The RNA quantities (500 ng per sample) used in our PCR experiments are now stated in the Methods section.

20. Smaller data sets within this study should be statistically analysed using non-parametric tests. Where this has not been done, I would suggest including a comment on normality within the methods.

Thank you for the insightful comment regarding the statistical analysis of smaller data sets. We fully acknowledge the importance of using non-parametric tests when the assumptions of normality may not be met. In response to the reviewer's suggestion, we reanalyzed the data with small sample sizes (e.g., n=4) using non-parametric tests, specifically the Mann-Whitney U test for two-group comparisons and the Kruskal-Wallis test followed by Dunn's multiple comparisons test for multiple group analyses. We confirmed that the results are consistent with our original findings, confirming the robustness of our data. This information was added to the figure legends and Methods section.

21. Can you include more detail on the ISG and cytotoxicity score calculation in the methods? A score of -0.5 for HD was not intuitive for me as a reader.

Thank you for commenting on and highlighting the need for additional details regarding the ISG and cytotoxicity score calculations. The scores were calculated using the AddModuleScore function in Seurat, a well-established package for single-cell RNA sequencing (scRNA-seq) analysis. This function calculates scores by averaging the expression levels of a predefined gene set and subtracting the average expression of a randomly selected set of control genes. As a result, the scores can range from negative to positive values, where negative scores indicate a lower relative expression of the gene set compared to the control set. The score of -0.5 for HD reflects such a calculation. We added this explanation to the Method section.

22. Note typo "Healthy" in Fig 1A.

Thank you for pointing out this typo, which was corrected in the revised Fig. 1a.

23. Certain papers have shown increased ISG expression within the kidneys of SLE and AAV patients. Neutrophils and monocytes have also been shown to be major sources of these transcripts within SLE patients, which does not agree with Fig. 6 within this manuscript that only noted DC/macrophages as the source of ISG expression within kidney samples. Please include this in your discussion.

<https://www.ncbi.nlm.nih.gov/pmc/articles/PMC8318865/> - Chen et al. eBioMedicine 2021

<https://www.ncbi.nlm.nih.gov/pmc/articles/PMC2760083/pdf/nihms142880.pdf> - Kessenbrock et al. Nat Med 2009

We agree with the reviewer. The findings in Fig.6 may be specific to the murine cGN model. The publications are now included in the reference list, and this point is discussed (please see page 17).

24. Please add a section within the discussion as to possible reasons why there is no upregulation of granzyme expression in blood RNAseq data (Fig.1) versus cytotoxicity in kidney (Fig 7/8).

Thank you for this insightful comment. We speculate the increased cytotoxicity of renal ISG-T cells compared to their blood counterparts may be due to the higher exposure of kidney ISG-T cells to interferon type I (IFN-I). This increased exposure could drive their cytotoxic function, causing higher granzyme expression in the kidney tissue. This hypothesis is included in the Discussion section (please see page 16).

25. It is noted that the authors only use male mice for their *in vivo* models of cGN (the nephrotoxic nephritis model), which in the context of the cGN model produces a more severe form of disease. Ideally moving forward, and for more translational relevance given the significant sex differences in the disease the authors focus on, I would strongly suggest the authors include female mice in future work.

Thank you for your valuable feedback and suggestions of including female mice in our *in-vivo* experiments. We acknowledge the importance of considering sex differences for greater translational relevance. For our future experiments, we will include female mice.

Reviewer: 4

Remarks to the Author:

We appreciate the reviewer's co-reviewing.

Reviewer #1:

Remarks to the Author:

Congratulations to the authors for this successful, intensive revision of the manuscript and I am happy with the additional experiments and changes. I have no further questions or comments.

Thank you!

Reviewer: 2

Remarks to the Author:

I appreciate the efforts made by the authors in revising the manuscript based on my previous comments. The revised version has improved in several aspects. However, there are still some limitations that need to be addressed to further substantiate the authors' conclusions. Below are my further comments and suggestions for additional refinements.

We sincerely thank the reviewer for the thoughtful comments and constructive suggestions. In response to the feedback from both the reviewer and the editor, we have further revised the manuscript to address the remaining concerns and improve the overall clarity and robustness of our conclusions.

1. In the part of 'Introduction', the author describes CTLs as mature cells that develop following T cell activation. However, the analysis results suggest that ISG-T cells possess cytotoxic functions despite being immature. This appears to be a contradiction. Could the author clarify this inconsistency?

We thank the reviewer for pointing out this inconsistency. We removed the term "mature" in the Introduction.

2. The authors propose that IFN-I induces IRF7 expression, however, the mechanism by which IRF7 regulates GZMB expression remains insufficiently supported by evidence. Given that STAT1 is also highly expressed in ISG-T cells, what was the rationale for prioritizing IRF7 for validation? Furthermore, IRF7 overexpression leads to an upregulation of STAT1—does IRF7 modulate ISG cytotoxicity through STAT1, or does it directly regulate GZMB expression?

We focused on IRF7 due to its highest expression level in the ISG-T cell population. We agree with the reviewer that STAT1 is also a key transcription factor involved in T cell differentiation, and the mechanism by which IRF7 regulates cytotoxicity remains an intriguing and important question. While our current data show that IRF7 overexpression leads to upregulation of STAT1, it remains unclear whether IRF7 modulates cytotoxicity through STAT1 or directly regulates GZMB expression. To address this, we are in the process of establishing techniques such as CUT&RUN and CUT&Tag to identify genes directly regulated by IRF7. However, we believe that these additional studies are beyond the current scope of our current manuscript.

3. Can the spatial transcriptomics data reveal the co-localization of ISGs and GZMB expression?

In the Xenium spatial transcriptomics analysis, some genes, including GZMB, were not detected as sensitively as in the scRNA-seq dataset. This is likely due to the imaging-based detection method used in Xenium, which differs from sequencing-based approaches in terms of sensitivity and dynamic range. As a result, we were unable to robustly assess the co-localization of ISG and GZMB expression in the spatial data.

4. Is the percentage in line 87 representing the mean or another metric? This should be clearly annotated.

The percentages reported throughout the manuscript, including in line 87, represent the mean values. We have clarified this point in the revised manuscript.

5. The designation of ISG-T as an immature state in Results 2 is not sufficiently substantiated. Although scCITE-seq data is provided, the absence of direct flow cytometry validation weakens the evidence supporting this classification. Moreover, the conclusions in the second and fourth sections of the results are redundant. Would it be advisable to consider merging them for improved clarity and conciseness?

While we only provided scCITE-seq data for patients with ANCA-GN, we also performed flow cytometry analysis on T cells from patients with SLE, which confirmed the immature phenotype of ISG-T cells based on their surface marker expression (Supplementary Fig. 3d-g). These findings support the classification of ISG-T cells as an immature state.

We also appreciate the suggestion of the reviewer to merge the second and fourth sections of the Results. After careful consideration, however, we believe that the current structure is better understandable compared to a merged section (due to the large amount and complexity of the data in a merged section).

6. In Supplementary Figure 4h, there is no evident positive correlation between the cytotoxicity score and the ISG score. Additionally, the R/R² values are not indicated.

We thank the reviewer for this advice. In the revised Supplementary Figure 4h, we have now included the R value to clarify the correlation. While the positive correlation is not strong and the R value is relatively low, the p-value remains < 0.05, indicating statistical significance. It is worth noting that in scRNA-seq analyses, correlation coefficients in scatter plots often appear smaller than in bulk transcriptome data due to inherent sensitivity limitations at the single-cell level.

Revised Supplementary Fig. 4h. Correlation analysis between the ISG score and cytotoxicity score in ISG-T cells. The Pearson correlation coefficient (r) and p-value are shown.

7. Figure 2 already presents the gene expression and cytotoxicity scores of ISG-T in the cGN model, yet Figure 3 also focuses on renal T cells in the cGN model. What is the rationale for separately reanalyzing CD4⁺ and CD8⁺ T cells?

In Figure 2, the ISG-T cell cluster includes a mixture of both CD4⁺ and CD8⁺ T cells. In Figure 3, we reanalyzed these populations separately to confirm that both CD4⁺ and CD8⁺ T cells can exhibit the ISG-T cell phenotype. Additionally, separating CD4⁺ and CD8⁺ T cells allowed us to assign a single naïve T cell cluster as the root of differentiation for each subset, thereby improving the interpretability of the trajectory.

8. Additionally, in Supplementary figure 11b, the ISG-CTL population in SLE exhibits a high pseudotime score, suggesting it represents a terminally differentiated, mature T cell subset. This appears inconsistent with the pseudotime trajectory analysis conducted separately in the cGN model. How can this discrepancy be explained?

We thank the reviewer for this insightful comment. While the ISG-CTL population in SLE also expresses interferon-stimulated genes (ISGs), their expression levels are significantly lower compared to those in the ISG-T cells identified in patients with ANCA-GN and in the murine cGN model (Fig. 1d). This suggests that the ISG-CTL population is distinct from the ISG-T cells we mainly focus on in our manuscript.

As correctly stated, the pseudotime analysis indicates that the ISG-CTL population may represent a terminally differentiated subset. However, due to the differences in ISG expression levels and functional characteristics, we do not think this finding contradicts our conclusion that ISG-T cells represent a more immature or early-stage population.

To avoid confusion, we have revised the manuscript and removed the phrase “another ISG-T cluster” in reference to the ISG-CTL population.

9. The pseudotime trajectory analysis does not explicitly indicate trajectory direction. How, then, can it be supposed that ISG-T represents an intermediate population in the differentiation process from naïve T cells to mature CTLs?(line 173-174)

We agree that pseudotime trajectory analysis does not explicitly indicate the direction of differentiation. In response to the reviewer’s comment, we have revised the sentence on lines 173–174 as follows:

Original:

The cytotoxic phenotype and low pseudotime score of ISG-T cells indicated that ISG-T cells potentially differentiate from naïve T cells into mature CTLs through IFN-I signaling.

Revised:

The expression of cytotoxic genes in kidney ISG-T cells suggests that IFN-I induces T cell cytotoxicity.

10. Does IRF7 knockout affect IFN-β-induced cytotoxicity and ISG expression in CD4⁺/CD8⁺ T cells? Additionally, was ISG expression assessed in IRF7 overexpression system?

We agree with the reviewer that the lack of IRF7 knockout experiments is a limitation of our study, and we have acknowledged this point in the Discussion section (Page 17). We attempted to obtain IRF7 knockout mice; however, they were not available. We also considered using a CRISPR/Cas9-mediated IRF7 knockout approach. However, this strategy requires T cell activation for gRNA transduction, and the activation process itself induces T cell cytotoxicity before the knockout is achieved, making the analysis of cytotoxicity and ISG expression challenging.

ISG expression was not assessed in the IRF7 overexpression experiments.

11. In Figure 6, is ISG expression selectively elevated in T cells within the cGN model, or is it also upregulated in other cell types? Similarly, is IRF7 expression restricted to CD4⁺ and CD8⁺ T cells, or is it observed in additional immune populations?

Furthermore, the initial patient analysis identified ISG-T in the peripheral blood of ANCA and SLE patients. Why, then, are GZMB⁺ CD4⁺/CD8⁺ T cells absent in both the peripheral blood and spleen of cGN mice?

It has been reported that IFN-I can influence a variety of immune cell types, including myeloid cells and B cells, in addition to T cells (PMID: 37821442, 29996101, 39375351, 38330141). In response to the reviewer’s question, we analyzed our scRNA-seq dataset of CD45⁺ cells isolated from the kidney at day 10 of the cGN model (the same dataset shown in Fig. 6a). In this analysis, we observed the expression of ISGs such as *Isg15*, *Isg20*, *Ifit1*, and *Irf7* across multiple leukocyte clusters, not limited to CD4⁺ and CD8⁺ T cells. These results indicate that IFN-I responses are broadly distributed among kidney-infiltrating leukocytes.

ISG expression in different leukocyte populations isolated from the inflamed kidney (at day 10 of the cGN model). UMAP plots showing cell type annotations and expression of selected ISGs across different leukocyte clusters.

ISG-T cell detection and GZMB detection

In our analysis of patients with AAV or SLE, ISG-T cells detected in the peripheral blood did not exhibit a cytotoxic phenotype (Fig. 1c; also shown below). In contrast, ISG-T cells isolated from the kidney of ANCA-GN patients showed a higher cytotoxicity score (Fig. 8d; also shown below). Based on these findings, we consider that ISG-T cells acquire cytotoxic potential primarily within inflamed tissues. As you noted, GZMB was not detectable in T cells from the peripheral blood or spleen in the cGN model. However, this observation is consistent with our findings in human samples and does not contradict the tissue-restricted cytotoxicity of ISG-T cells.

Fig. 1c ANCA dataset - Marker genes

SLE dataset - Marker genes

Fig. 8d

ISG-T cells gain cytotoxicity in inflamed kidney tissue. Left: Heatmaps (from Fig. 1c) showing marker gene expression in blood T cell clusters, including ISG-T cells, from patients with AAV or SLE. Right: Violin plots (from Fig. 8d) showing the ISG score and cytotoxicity score of ISG-T cells isolated from the blood and kidney of ANCA-GN patients.

12. Spatial transcriptomics data indicate that the cell type closest to ISG-T is pDC. What is the nature of the interaction between pDCs and ISG-T? Are there specific ligand-receptor interactions or cytokine signaling pathways mediating their crosstalk?

pDCs are well known as primary producers of IFN-I (PMID: 29559718, 25614319). Based on this, we speculate that pDCs may interact with ISG-T cells through IFN-I signaling. However, due to the low expression levels of IFN-I genes such as IFNA1 and IFNB1 (PMID: 28420733, 36434007, 19667093), these transcripts were not sensitively detected on the Xenium platform. As a result, we were unable to perform a meaningful ligand-receptor-based interactome analysis to further investigate this potential crosstalk.

Reviewer: 3

Remarks to the Author:

The authors have adequately addressed all my comments. Their changes to this paper in response to reviewers' comments significantly strengthens the insights into novel T cell functionality in autoimmune kidney diseases.

Thank you!